# A neuronal code for object representation and memory in the human amygdala and hippocampus

Runnan Cao [1] ✉, Peter Brunner[2], Puneeth N. Chakravarthula[1], Krista L. Wahlstrom[3], Cory Inman [3], Elliot H. Smith[4], Xin Li [5], Adam N. Mamelak [6], Nicholas J. Brandmeir[7], Ueli Rutishauser[6], Jon T. Willie [2,8] ✉ & Shuo Wang [1,2,8] ✉

How the brain encodes, recognizes, and memorizes general visual objects is a fundamental question in neuroscience. Here, we investigated the neural processes underlying visual object perception and memory by recording from 3173 single neurons in the human amygdala and hippocampus across four experiments. We employed both passive-viewing and recognition memory tasks involving a diverse range of naturalistic object stimuli. Our findings reveal a region-based feature code for general objects, where neurons exhibit receptive fields in the high-level visual feature space. This code can be validated by independent new stimuli and replicated across all experiments, including fixation-based analyses with large natural scenes. This region code explains the long-standing visual category selectivity, preferentially enhances memory of encoded stimuli, predicts memory performance, encodes image memorability, and exhibits intricate interplay with memory contexts. Together, region-based feature coding provides an important mechanism for visual object processing in the human brain.

A key challenge in cognitive neuroscience revolves around understanding the process of object recognition, a process that assigns unique identity labels to distinct visual forms. For this to happen, visual information must be transformed into retrievable memories. The human amygdala and hippocampus play a critical role in recognition memory[1–6], and neurons in the human amygdala and hippocampus signal whether stimuli are novel or familiar[7,8]. In particular, for years, neurons in the human amygdala and hippocampus have been associated with category-specific encoding of visual objects and faces[9–11], structured in a hierarchical fashion[12]. These category-selective concept neurons form highly selective (sparse) representations of objects[13–16] and are regarded as the building blocks of declarative memory[15–18].

However, it remains largely unknown how such sparse representations are formed in the amygdala and hippocampus, and how visual information processing is linked to memories.

Neurons in the higher visual cortex respond to complex visual stimuli such as faces and objects[19,20] and demonstrate feature-based coding of objects. According to this model, objects are represented across a broad and distributed population of neurons[21–24]. In a specific form of feature-based coding known as axis-based feature coding, visual cortical neurons or voxels parametrically correlate with visual features along specific axes in feature space[20,25–29]. Neurons in the amygdala and hippocampus likely receive this highly processed visual information as input and form high-level visual representations of

[1]Department of Radiology, Washington University in St. Louis, St. Louis, MO, USA. [2]Department of Neurosurgery, Washington University in St. Louis, St. Louis, MO, USA. [3]Department of Psychology, University of Utah, Salt Lake City, UT, USA. [4]Department of Neurosurgery, University of Utah, Salt Lake City, UT, USA. [5]Department of Computer Science, University at Albany, Albany, NY, USA. [6]Department of Neurosurgery, Cedars-Sinai Medical Center, Los Angeles, CA, USA. [7]Department of Neurosurgery, West Virginia University, Morgantown, WV, USA. [8]These authors jointly supervised this work: Jon T Willie, Shuo Wang. ✉e-mail: r.cao@wustl.edu; jontwillie@wustl.edu; shuowang@wustl.edu

stimuli[18]. Such category-selective neurons have been hypothesized to represent semantic memories[30], which in turn form the foundation of declarative memories[16]. In our recent study, we have shown that a subset of human amygdala and hippocampal neurons exhibit a novel region-based feature code for faces[31]. Specifically, based on visual features encoded in the higher visual cortex, these neurons exhibit receptive fields in the high-level visual feature space, making them responsive to stimuli that fall into the coding/tuning region (i.e., receptive field), thereby providing a bridge between the visual feature coding and category coding mechanisms.

In this study, we set out to understand how visual information processing is linked to object category representations and recognition memory. We utilized both passive-viewing visual working memory tasks and recognition memory tasks involving a diverse range of naturalistic object stimuli, and we applied an object space approach, which involved using high-level visual features extracted from individual images by a deep neural network (DNN), without assuming categorical membership for each image. We first generalized and cross-validated the region-based feature code through four experiments (two passive-viewing, one phased recognition memory, and one continuous recognition memory), including fixation-based analysis for visual objects from large natural scene images. Based on this general region-based feature code, we tested two major hypotheses: (1) region-based feature code can explain visual category selectivity, and (2) it predicts memory performance (i.e., stimuli in the coding regions are preferentially encoded into memory) and image memorability. Finally, we explored the flexibility of this code in different memory contexts. By investigating these questions, our results provide a critical link between visual information processing and memory at the single-neuron level in humans.

## Results

### Region-based encoding of visual objects

We recorded from 1204 neurons in the amygdala and hippocampus (together referred to as the medial temporal lobe [MTL]) of 15 neurosurgical patients (5 male; 28 sessions in total; Supplementary Table 1) while they performed a one-back task ("Methods" section; accuracy = 81.42% ± 20.93% [mean ± SD across sessions]). Participants viewed 500 natural object pictures of 50 categories from the ImageNet dataset (10 different pictures per object category). 874 neurons had an overall mean firing rate greater than 0.15 Hz and we restricted our analysis to this subset of neurons, which included 477 neurons from the amygdala, 198 neurons from the anterior hippocampus, and 199 neurons from the posterior hippocampus (Supplementary Table 1; see Supplementary Fig. 1a–h for spike sorting metrics and Supplementary Fig. 1i for recording locations).

We have recently demonstrated that neurons in the human amygdala and hippocampus exhibit region-based feature coding for faces, i.e., these neurons respond to faces that fall into a specific region of the feature space[31]. In this study, we explored whether this region-based feature code in the high-level visual feature space also extends to objects in general. Similar to the construction of a face feature space[31], we extracted visual features from the object pictures presented to the patients using the last convolutional layer (Res5c) of a pre-trained deep neural network (DNN) known as ResNet, which is trained to recognize objects (similar results were derived using other object recognition DNNs, such as the AlexNet). We further reduced the dimensionality of the DNN features to construct a two-dimensional feature space using t-distributed stochastic neighbor embedding (t-SNE) (Fig. 1a, b; similar results could be derived when we used UMAP or PCA for dimension reduction; see "Methods" section). The feature space exhibited an organized structure, with objects of the same category clustered together. Feature Dimension 1 primarily represented the transition from artificial to natural objects (consistent with ref. 20), while Feature Dimension 2 mostly

captured variations in object size and the shift from indoor to outdoor objects.

We then projected the response of a given neuron to each object onto the feature space by multiplying the response magnitude of each object by its corresponding location in the feature space, resulting in a response-weighted 2D feature map (Fig. 1a, b middle). This revealed that a subset of MTL neurons was selective for objects clustered together in the feature space and tuned to specific regions within it (Fig. 1a, b middle), suggesting that these neurons responded to objects sharing similar visual features. We refer to this category of MTL neurons as feature neurons. To formally quantify the tuning of feature neurons (see "Methods" section; Supplementary Table 1), we estimated a continuous spike density map in the 2D feature space (Fig. 1a, b upper right) by smoothing the discrete firing rate map (Fig. 1a, b middle) using a 2D Gaussian kernel and used a permutation test (1000 runs; Fig. 1a, b lower right) to identify the region(s) that had a significantly higher spike density above chance (red/cyan outlines in Fig. 1a, b; significant pixels were selected with permutation $P < 0.01$ and cluster size thresholds; similar results were derived with further cluster correction across adjacent significant pixels[32]). This region indicates the part of the feature space to which a neuron was tuned. We refer to such coding of the feature neurons as *region-based feature coding* because they coded a certain region in the feature space. At the population level, we observed a significant number of feature neurons ($n = 89$, 10.18%, binomial test $P = 1.60 \times 10^{-10}$; see Supplementary Table 1 for a breakdown of amygdala and hippocampal neurons). The number of object categories (5.45 ± 2.56 [mean ± SD across neurons]) and objects (21.52 ± 9.72 [mean ± SD across neurons]) covered by the tuning region of feature neurons indicated the size of the receptive field (in feature space) of these feature neurons. The tuning region of each feature neuron covered approximately 1.75-9.64% of the feature space and the total observed feature neuron population covered approximately 63.43% of the feature space (Fig. 1c).

Lastly, it is worth noting that although the quantifications were in the t-SNE space, we replicated our results in the full dimensional space of the DNN (Fig. 1d; two-tailed paired t-test: $t(81) = -3.43$, $P = 0.0009$, $d = 0.37$, 95% CI = [−0.0097, −0.036]). Moreover, we found that only 3.20% (below the 5% chance level) of the amygdala and hippocampal neurons exhibited axis-based feature coding (i.e., neurons encoding a linear combination of DNN visual features; see "Methods" section for details), consistent with our prior report examining neuronal responses to faces[31]. Therefore, in contrast to the macaque inferotemporal (IT) cortex[20,26–28], feature-based coding in the human MTL was primarily region-based rather than axis-based.

Together, our results suggest that amygdala and hippocampal neurons exhibit region-based feature coding for general objects.

### Region-based feature coding is a more comprehensive mechanism for object coding

Category-specific encoding of objects is a hallmark for MTL neurons[9–12,33,34]. We next investigated whether category-specific encoding of objects could be explained within the feature coding framework. It is worth noting that our analyses did not require category coding and feature coding to be separate processes or de-correlated; instead, we explored a more comprehensive explanation for the long-standing phenomenon of category selectivity.

We observed that MTL neurons encoded category memberships of visual objects (Fig. 1a, b). To classify *category-selective neurons*, we first used a one-way (1 × 50) ANOVA to identify neurons with a significantly different response across object categories ($P < 0.05$) in a window 250–1250 ms following stimulus onset. We imposed a second criterion to identify to which categories a neuron was selectively responding (selected categories): the neural response to such a category was required to be at least 1.5 standard deviations (SD) above the mean neural response during baseline (−500 to 0 ms relative to

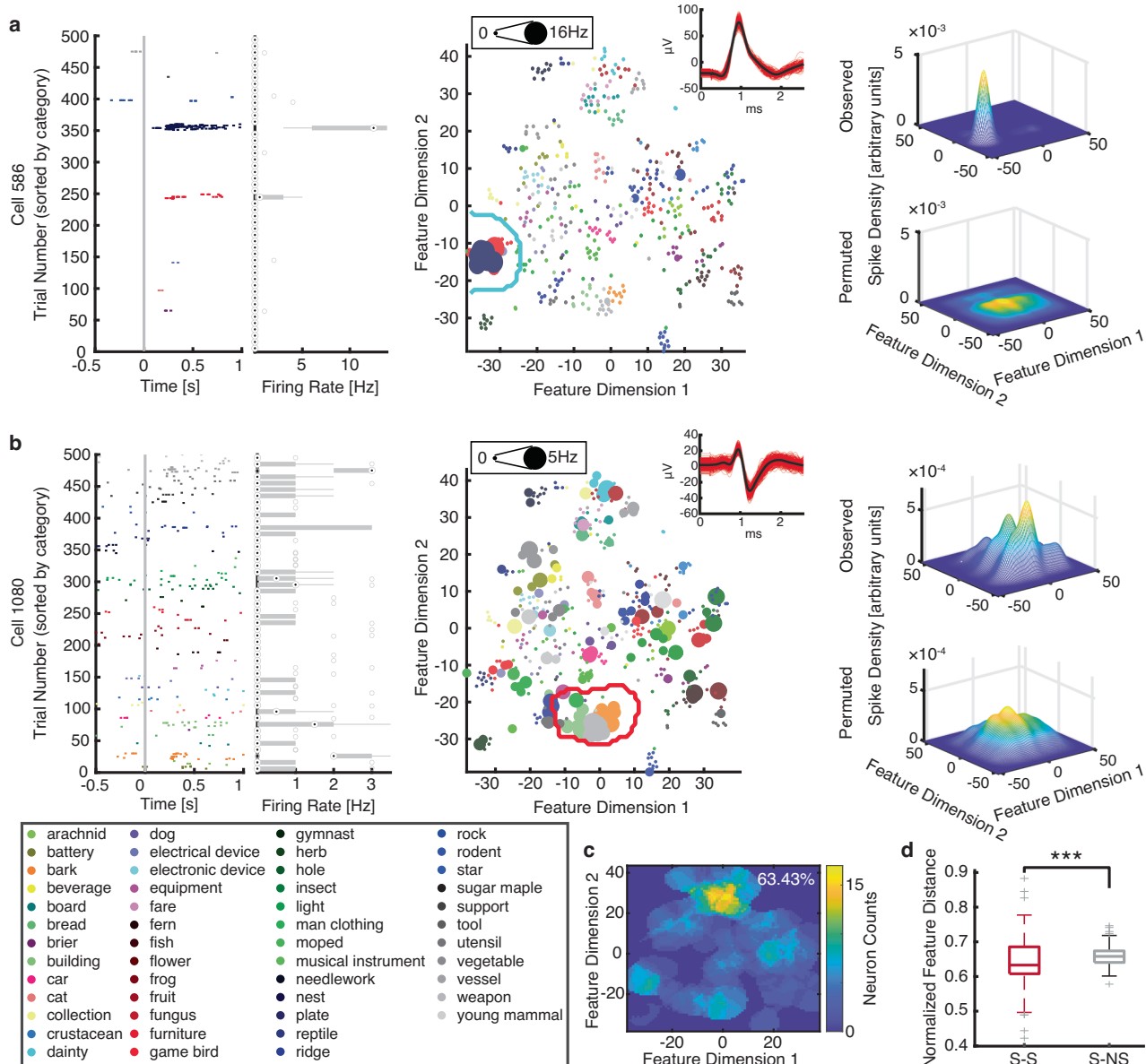

**Fig. 1 | Feature-based neuronal coding of general objects. a, b** Two example neurons that encoded visually similar object categories (i.e., feature MC neurons). (Left) Neuronal responses to 500 objects (50 object categories). Trials are aligned to stimulus onset (gray line) and are grouped by individual object categories. Error bars denote ±SEM across objects. (Middle) Projection of the firing rate onto the feature space. Each color represents a different object category. The size of the dot indicates the firing rate. The inset shows the waveform of the unit. (Right) Estimate of the spike density in the feature space. By comparing observed (upper) vs. permuted (lower) responses, we could identify a region where the observed neuronal response was significantly higher in the feature space. This region was defined as the tuning region of a neuron (delineated by the red/cyan outlines). **c** The

aggregated tuning regions of the feature neuron population. The color bar shows the number of neurons with tuning regions in a given area. The number on the upper right corner shows the percentage of feature space covered by at least one neuron. **d** Comparison of DNN full feature distance in feature MC neurons between selective-selective (S-S) object category pairs (i.e., neurons were selective to both categories; shown in red; *n* = 82) vs. selective-non-selective (S-NS) object category pairs (i.e., neurons were selective to only one of the categories; shown in gray; *n* = 82) using a two-tailed paired *t*-test (see "Methods" section). ****P* < 0.001. Each box shows the median (central mark), 25th and 75th percentiles (box edges), whiskers (non-outlier extremes), and individual outliers. Source data are provided as a Source Data file.

stimulus onset) to all categories. Using this established procedure to select category-selective neurons[35,36], we found that 121 neurons satisfied both criteria (13.84%, binomial *P* < 10^−20; Fig. 2a; see Supplementary Fig. 2a, f for a breakdown of the amygdala and hippocampal neurons), and of these category neurons, 39 responded to a single category only (referred to here as *single-category [SC] neurons*) and the remaining 82 neurons each responded to multiple categories (referred to here as *multiple-category [MC] neurons*). On average, MC neurons encoded 5.37 ± 4.21 visual categories. In particular, we found that 45/82 MC neurons (54.88%; Fig. 2a) responded to object categories clustered

within the same region in the feature space, suggesting that these neurons encoded object categories with similar visual features. We referred to these neurons as *feature MC neurons*. On the other hand, the remaining MC neurons encoded object categories distributed in separate locations in the feature space that were not part of the same region; and we referred to these neurons as *non-feature MC neurons* (Fig. 2a). In summary, SC neurons encoded a single narrow peak in the feature space encompassing objects from a single category, feature MC neurons encoded a single wide peak in the feature space encompassing objects from multiple categories, and non-feature MC neurons

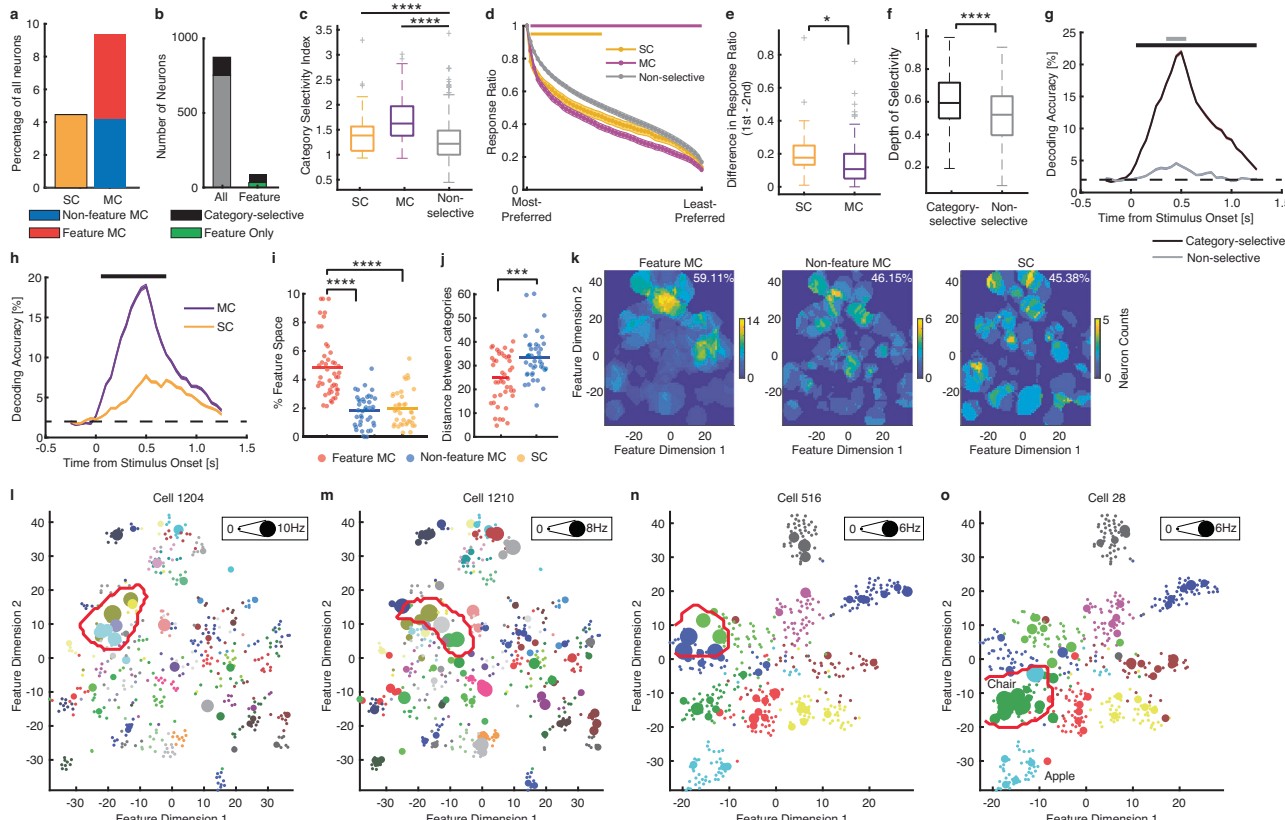

**Fig. 2 | Feature coding for category-selective neurons. a** Percentage of single-category (SC; shown in yellow) and multiple-category (MC) neurons in the entire neuronal population. Stacked bar shows MC neurons that encoded visually similar objects (i.e., feature MC neurons; red) or not (i.e., non-feature MC neurons; blue). **b** The number of category-selective neurons in the whole population (left) and among feature neurons (right). Black: the number of category-selective neurons ($n = 121$ for the whole population and $n = 56$ for feature neurons). Green: the number of non-category-selective feature neurons ($n = 33$). Gray: the number of non-category-selective neurons ($n = 753$). **c** Category selectivity index. Both SC neurons ($n = 39$; shown in yellow) and MC neurons ($n = 82$; combining both feature MC and non-feature MC neurons; shown in magenta) had a significantly higher category selectivity index than non-category-selective neurons ($n = 753$). Asterisks indicate a significant difference using a two-tailed two-sample $t$-test. *$P < 0.05$, **$P < 0.01$, ***$P < 0.001$, and ****$P < 0.0001$. **d** Ordered average responses from the most-preferred to the least-preferred object category. Non-category-selective neurons are shown for comparison purposes. Responses were normalized by the response to the most-preferred category. Shaded areas denote ±SEM across neurons. The top bars indicate significant differences between SC/MC and non-category-selective neurons (two-tailed unpaired $t$-test: $P < 0.05$, corrected by FDR[67] for $Q < 0.05$). **e** Difference in response ratio between the most-preferred and second most-preferred object categories (SC: $n = 39$; MC: $n = 82$). **f** Depth of selectivity (DOS) index. Category-selective neurons ($n = 121$) had a significantly higher DOS index than non-category-selective neurons ($n = 753$). Each box shows the median (central mark), 25th and 75th

percentiles (box edges), whiskers (non-outlier extremes), and individual outliers. **g, h** Population decoding of object category. Shaded area denotes ±SEM across bootstraps. The horizontal dotted lines indicate the chance level. **g** Decoding performance was primarily driven by category-selective neurons (black). The top bars illustrate the time points with a significant above-chance decoding performance (bootstrap, $P < 0.05$, corrected by FDR for $Q < 0.05$). **h** MC neurons had a significantly better decoding performance than SC neurons because the encoding by MC neurons was less sparse. The top bar illustrates the time points with a significant difference between MC and SC neurons (bootstrap, $P < 0.05$, corrected by FDR for $Q < 0.05$). **i** Percentage of feature space covered by tuning regions of category-selective neurons. Note that here we did not apply the threshold for minimum cluster size for SC and non-feature MC neurons in order to compare across different categories of category-selective neurons. **j** Normalized distance between MC neurons' selected categories in the feature space. Euclidean distance was normalized by the maximum distance (i.e., diagonal line) of the feature space. Each dot denotes a neuron and the horizontal bar denotes the mean. Asterisks indicate a significant difference between feature MC neurons and non-feature MC neurons using a two-tailed two-sample $t$-test. *$P < 0.05$, **$P < 0.01$, and ***$P < 0.001$. **k** The aggregated tuning regions of the feature neuron population. Numbers in the density maps show the percentage of feature space covered by the tuning regions of the total observed feature neurons. **l–o** Example feature neurons that did not depend on category selectivity. Legend conventions as in Fig. 1. **l, m** ImageNet stimuli. **n, o** Microsoft COCO stimuli. Source data are provided as a Source Data file.

encoded multiple separate peaks in the feature space with each peak encompassing objects from a different category. Notably, feature neurons had a substantially higher proportion (56/89; 62.92%) of category-selective neurons compared to the entire neuronal population (121/874; 13.84%; $\chi^2$-test: $P < 10^{-20}$; Fig. 2b; see Supplementary Fig. 2b, g for a breakdown of the amygdala and hippocampal neurons), suggesting that feature neurons were more likely to be category-selective neurons. We next analyzed each group of category-selective neurons within the framework of object feature space.

We first analyzed category selectivity for each neuron using a category selectivity index ($d'$ between the most-preferred and least-preferred categories; Fig. 2c; SC vs. non-selective: $t(790) = 7.60$,

$P = 8.64 \times 10^{-14}$, $d = 1.25$, 95% CI = [0.36, 0.61]; MC vs. non-selective: $t(833) = 7.22$, $P = 1.16 \times 10^{-12}$, $d = 0.84$, 95% CI = [0.24, 0.42]) and ordered responses from the most-preferred to the least-preferred category (Fig. 2d). As expected, compared to the non-category-selective neurons, MC neurons showed broadly significantly decreasing relative responses from the most-preferred to the least-preferred category, whereas SC neurons showed steeper changes from the most-preferred to the least-preferred category (Fig. 2d). This selectivity was also evident in the larger decrease of firing rate between the most-preferred and the second most-preferred stimuli in SC neurons compared to MC neurons (Fig. 2e; two-tailed two-sample $t$-test: $t(119) = 2.24$, $P = 0.03$, $d = 0.43$, 95% CI = [0.008, 0.12]). We further

confirmed these results using a depth of selectivity (DOS) index (Fig. 2f; $t(872) = 5.96$, $P = 3.66 \times 10^{-9}$, $d = 0.58$, 95% CI = [0.06, 0.12]). In addition, single-trial population decoding revealed that MC neurons differentiated between more stimuli than SC neurons (Fig. 2g, h; note that similar results were derived when we matched the number of neurons between groups). The tuning region of an individual feature MC neuron covered 2.15–9.64% of the 2D feature space (Fig. 2i). In contrast, the response of an individual SC or non-feature MC neuron covered a significantly smaller region in the feature space (Fig. 2i; two-tailed two-sample $t$-test: $P < 0.0001$ for both comparisons). This result was as expected because the object categories (and thus the tuning regions) encoded by non-feature MC neurons were not contiguous with each other and were further apart (Fig. 2j). As a whole, the population of feature MC neurons sampled covered various areas of the 2D feature space (Fig. 2k; some areas were encoded by multiple neurons), suggesting that these neurons encoded a variety of visual features. It is worth noting that SC units and MC units had a similar spike sorting isolation distance (Supplementary Fig. 1h; $t(82) = 0.16$, $P = 0.87$, $d = 0.04$, 95% CI = [−0.27, 0.32]), suggesting that MC units were not more likely to be multi-units consisting of several SC units.

Importantly, feature neurons could account for neural responses that were not explained by category selectivity. Specifically, there were feature neurons that were not category-selective neurons (Fig. 2b). First, these feature neurons encoded a subset of images from multiple categories, as long as the encoded objects shared similar visual features (Fig. 2l, m). Second, feature neurons might not respond to all objects from the same category (Fig. 2l–n). This was especially evident in the case of the Microsoft COCO dataset (see the section below for details), which included 50 images per object category (Fig. 2n). Third, interestingly, in cases of "misclassification" (i.e., when an object image was clustered based on visual features but not categorical membership), the neural response followed the feature rather than the category. For example, an indoor scene image with pictures of apples had a categorical membership of "apple" but was clustered together with indoor scene images of "chair" rather than other "apple" images (Fig. 2o). A feature neuron responding to "chairs" also had an elevated response to this "apple" image, but not to other "apple" images (Fig. 2o). Therefore, feature coding, rather than category selectivity, could better explain these data.

Together, these results suggest that region-based feature coding can serve as a more comprehensive mechanism for object coding. The classical category-specific neural response to objects can largely be explained within the framework of region-based feature coding (see "Discussion" section).

## Validation of region-based feature coding

We conducted an additional experiment to validate region-based feature coding using different stimuli. We recorded 453 neurons in 13 patients (18 sessions; firing rate >0.15 Hz; accuracy = 90.57% ± 12.25% [mean ± SD across sessions]) using object stimuli from the Microsoft COCO dataset. We applied the DNN AlexNet to extract features and construct the feature space. Again, we found that 61 neurons (13.5%; binomial $P = 1.19 \times 10^{-12}$) exhibited region-based feature coding in this experiment (Fig. 3a, b; see Supplementary Fig. 2c, d, h, i for a breakdown of the amygdala and hippocampal neurons).

Notably, a subset of neurons (152/453) were recorded using both the COCO and ImageNet stimuli. Therefore, we were able to directly investigate the generalizability of feature tuning between these two tasks. In the common feature space for the ImageNet and COCO stimuli (see Supplementary Fig. 3a, b for stability of stimulus representations across different constructions of feature spaces), the tuning regions of 22 feature neurons selected using the COCO stimuli encompassed the ImageNet stimuli. We predicted that the ImageNet stimuli located in the tuning regions of COCO feature neurons would elicit stronger responses compared to the other ImageNet stimuli

outside those tuning regions. Our results confirmed this hypothesis (right-tailed paired $t$-test: $t(21) = 1.81$, $P = 0.04$, $d = 0.16$, 95% CI = [0.02, ∞]; see Fig. 3c for an example and Fig. 3d for group results; see Supplementary Fig. 3c-f for control analyses with training and testing stimuli from the same dataset [Supplementary Fig. 3c, d for the ImageNet stimuli and Supplementary Fig. 3e, f for the COCO stimuli]), suggesting that region-based feature tuning generalized between different image sets.

Lastly, we further validated region-based feature coding with fixation-based analysis when patients viewed large natural scene images (Supplementary Information; Supplementary Fig. 4). These results validate region-based feature coding across independent experiments and in multiple contexts, including active vision.

## Region-based feature coding predicts recognition memory

Above, we demonstrated that region-based feature coding can serve as a more comprehensive framework to explain the classical category-specific neural response to objects in the human MTL. Category-selective sparse-coding neurons are considered the building blocks of declarative memory[15–18]. However, it remains unclear whether region-coding feature neurons, which not only explain category selectivity but also exhibit sparse-coding properties, are linked to aspects of declarative memory, particularly when comparing remembered vs. forgotten stimuli, as shown in prior research[37]. Moreover, some stimuli are more easily remembered due to their inherent perceptual features or saliency. Given that region-based feature coding is fundamentally a perceptual process, we hypothesize that stimuli encoded through this mechanism (in-region stimuli) may be preferentially stored in memory. To address these questions, we next investigated the contribution of region-based coding in feature neurons to memory performance using a phased recognition memory task[38] (Fig. 4a; see "Methods" section).

We included 1162 MTL neurons (666 from the amygdala and 496 from the hippocampus) recorded from 40 patients (57 sessions) with an overall mean firing rate greater than 0.15 Hz during both the learning and recognition task phases (note that neurons were sorted across task phases and exhibited consistent responses throughout). We constructed feature spaces for each task variant using stimuli combined from both the learning and recognition phases ("Methods" section; see Supplementary Fig. 5a, b for examples). We identified 116 feature neurons (9.98%; binomial $P = 1.45 \times 10^{-12}$) that exhibited region-based feature coding during the learning phase (see Fig. 4b–d for examples). Below, we investigated whether region-based feature coding during learning predicts subsequent memory performance during recognition.

Specifically, we compared recognition performance for stimuli that fell within vs. outside feature neurons' tuning regions identified during the learning phase. First, across the 65 feature neurons from sessions with hit rates exceeding 60% (note that 17 low-performance sessions were excluded to ensure reliability of the data), the hit rate (i.e., the proportion of trials correctly recognizing "old" stimuli) for in-region stimuli was significantly higher than that for out-region stimuli (Fig. 4e; $t(64) = 3.11$, $P = 0.0028$, $d = 0.46$, 95% CI = [0.02, 0.09]), suggesting that stimuli within feature neurons' tuning regions were better remembered. Furthermore, the confidence in memory for in-region stimuli was significantly higher than that for out-region stimuli (Fig. 4f; $t(64) = 2.28$, $P = 0.026$, $d = 0.33$, 95% CI = [0.24, 0.36]). We obtained similar results when we aggregated the tuning regions of feature neurons for each session and compared across sessions (Fig. 4g, h; hit rate: $t(27) = 2.83$, $P = 0.0086$, $d = 0.55$, 95% CI = [0.02, 0.12]; response/confidence level: $t(27) = 2.62$, $P = 0.014$, $d = 0.55$, 95% CI = [0.07, 0.55]).

Together, our results have demonstrated that stimuli within feature neurons' tuning regions are not only better remembered but also associated with greater memory strength, suggesting a link between visual feature coding and memory.

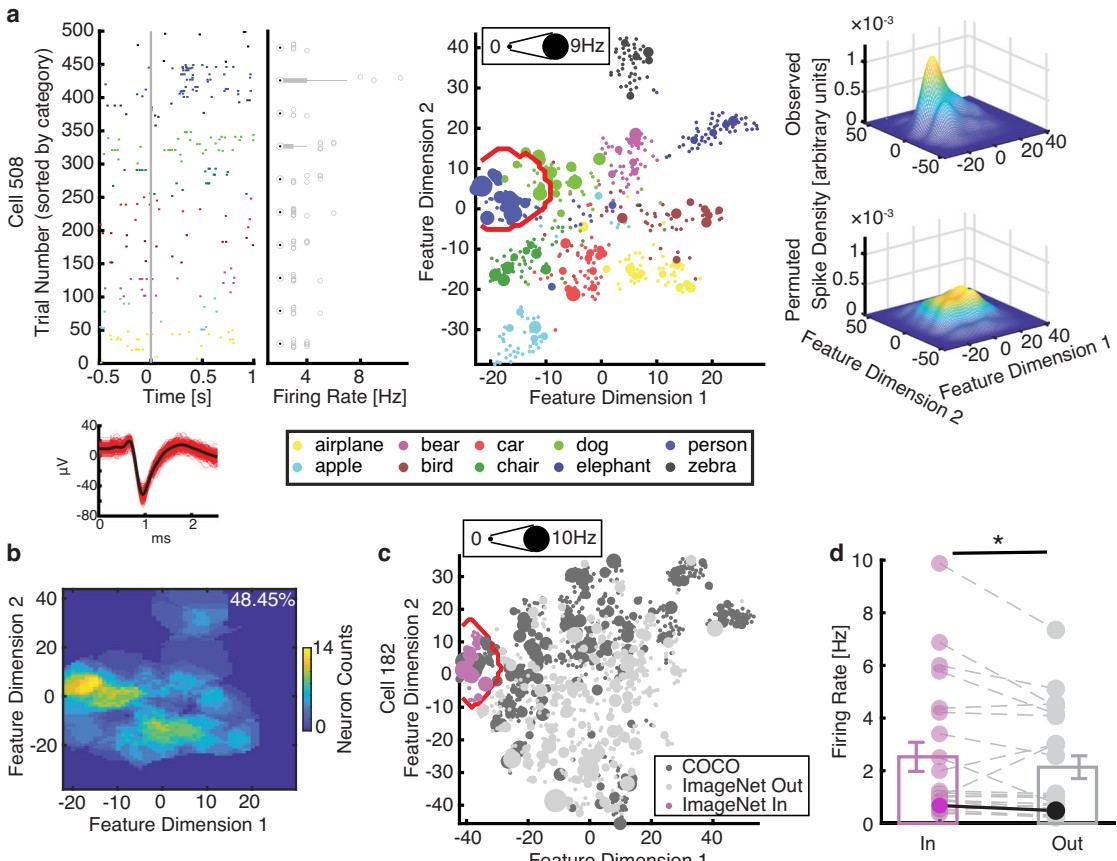

**Fig. 3 | Validation of region-based feature coding using Microsoft COCO stimuli. a** An example neuron demonstrating region-based feature coding. **b** The aggregated tuning regions of the neuronal population. Legend conventions as in Fig. 1. **c** An example COCO feature neuron showing elevated responses for ImageNet stimuli falling in its tuning region. The feature space was constructed for combined ImageNet and COCO stimuli. The size of the dot indicates the firing rate. The red outline delineates the tuning region of the neuron (identified by the COCO stimuli). Black: objects from the COCO stimuli. Gray: objects from the ImageNet stimuli. Magenta: ImageNet stimuli falling in the tuning region of the COCO feature neuron. **d** Population results comparing neuronal response to ImageNet stimuli falling in vs. out of the tuning region (*n* = 22). Each dot represents a neuron. Error bars denote ±SEM across neurons. Asterisks indicate a significant difference between the In vs. Out responses using a right-tailed paired *t*-test (*P* < 0.05). The example neuron shown in (**c**) is highlighted in a darker color. Source data are provided as a Source Data file.

## Feature neurons encode image memorability

Intrinsic image memorability—the likelihood that an image will be remembered by individuals after viewing it—consistently influences memory behavior across observers[39,40]. Research has shown that certain visual properties make some images more memorable than others, regardless of personal significance or familiarity[40]. Factors such as distinctiveness, emotional impact, and meaningful content contribute to this memorability. Since memorability may not represent the same neurocognitive processes as explicitly reported memory performance[39], and given that encoding memorability actively engages the human MTL[41], we next investigated whether feature neurons were involved in encoding general image memorability, in addition to predicting memory performance. Advanced computational models, such as ResMem[40], have been developed to predict how memorable an image is based on its visual features. We employed this well-established pre-trained DNN implementation of memorability to calculate the memorability score for each image ("Methods" section; see Fig. 5a for examples and Fig. 5b for the distribution of memorability scores for all stimuli).

We first confirmed that memorability scores predicted behavioral performance. Compared to low-memorability stimuli (i.e., stimuli with the bottom 30% memorability scores), high-memorability stimuli (i.e., stimuli with the top 30% memorability scores) resulted in a significantly higher recognition accuracy (including both hits and correct rejections; Fig. 5c; $t(60) = 8.51$, $P = 6.70 \times 10^{-12}$, $d = 1.24$, 95% CI = [0.09, 0.14]), greater confidence in memory decisions (Fig. 5d; $t(60) = 4.35$, $P = 5.29 \times 10^{-5}$, $d = 0.34$, 95% CI = [0.08, 0.21]), and faster responses (Fig. 5e; $t(60) = 4.03$, $P = 1.61 \times 10^{-4}$, $d = 0.21$, 95% CI = [−153.81, −51.71]). In particular, with stimuli that appeared in both the learning and recognition phases, we observed a higher hit rate (i.e., correctly recognizing "old" stimuli) for high-memorability stimuli (Fig. 5f; $t(31) = 3.01$, $P = 0.005$, $d = 0.55$, 95% CI = [0.03, 0.15]). Therefore, image memorability scores were associated with actual memory behaviors.

We next investigated whether feature neurons were more likely to encode image memorability during the learning phase (see Supplementary Information for analysis of the recognition phase; Supplementary Fig. 5). First, we observed a significant population of MTL neurons that correlated with levels of image memorability (Pearson correlation between firing rate and memorability score, $P < 0.05$; 104/1162, 8.95%, binomial $P = 7.44 \times 10^{-9}$; 47 neurons increased firing rate [Fig. 5g, i] and 57 neurons decreased firing rate [Fig. 5h, j] for higher memorability). Importantly, feature neurons had a significantly higher proportion encoding image memorability (31/116, 26.72%, binomial $P = 1.44 \times 10^{-15}$) compared to all recorded neurons ($\chi^2$-test between feature [$n_{Feature\&Memorability}$ / $n_{Feature}$] and all [$n_{Memorability}$ / $n_{All}$]: $P < 10^{-20}$; Fig. 5k), suggesting that feature neurons were more involved in encoding image memorability compared to non-feature neurons. Furthermore, 56 out of 116 feature neurons significantly differentiated

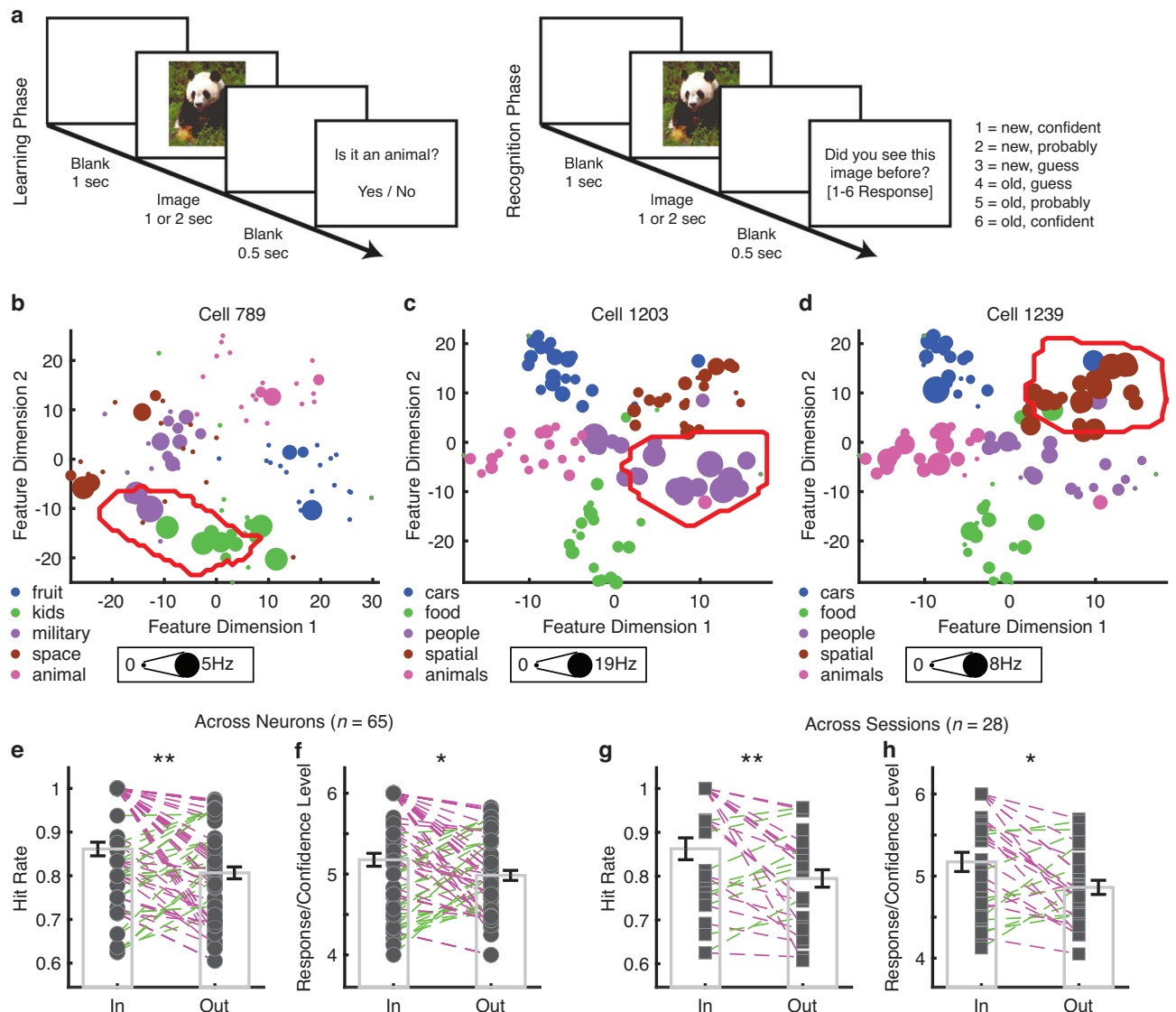

**Fig. 4 | Region-based feature coding predicts memory. a** Phased recognition memory task. This task comprised a learning phase where participants viewed 100 new images and determined whether each one contained an animal. The recognition test phase presented both new and old images, and participants indicated whether they had seen each image before, reporting their confidence levels. Stimuli were displayed on the screen for either 1 or 2 s. Images are from Faraut et al[38]. Data from: Dataset of human medial temporal lobe single-neuron activity during declarative memory encoding and recognition [Dataset]. Dryad. https://doi.org/10.5061/dryad.46st5. **b**–**d** Example feature neurons identified from the learning phase. Legend conventions as in Fig. 1. **e**–**h** Comparison of recognition performance for stimuli that fell within vs. outside feature neurons' tuning regions identified during the learning phase. **e**, **g** Hit rate (i.e., the proportion of trials correctly recognizing "old" stimuli). **f**, **h** Response/confidence level (see **a**). 4: least confidence in the decision to indicate "old". 6: most confidence in the decision to indicate "old". **e**, **f** Comparison across feature neurons ($n = 65$). Each circle represents a neuron, and error bars denote ±SEM across neurons. Magenta: In >Out. Green: In <Out. **g**, **h** Comparison across sessions ($n = 28$). Tuning regions of feature neurons from each session were aggregated. Each square represents a session, and error bars denote ±SEM across sessions. Asterisks indicate a significant difference between in-region vs. out-region stimuli using a two-tailed paired $t$-test. *$P < 0.05$, and **$P < 0.01$. Source data are provided as a Source Data file.

memorability scores between in-region and out-region stimuli, suggesting that feature neurons' tuning regions differentiated image memorability. It is worth noting that, based on previous memory studies[42], we expected to observe a mix of two populations of MTL neurons: one exhibiting a higher firing rate for more memorable stimuli and the other exhibiting a higher firing rate for less memorable stimuli. Both populations were considered to carry information and encode image memorability.

While, at the population level, in-region stimuli had a significantly lower memorability score compared to out-region stimuli (two-tailed paired $t$-test: $t(115) = 3.72$, $P = 0.0003$, $d = 0.51$, 95% CI = [0.014, 0.046]; note that 50/116 neurons had a higher memorability

score for in-region stimuli), stimuli with lower memorability scores could still be better remembered (Fig. 4e–g) as long as they fell within the tuning regions of feature neurons. This result also suggests that feature neurons may play a role in enhancing the memory of stimuli within their regions. Indeed, we found that in-region stimuli had a significantly higher hit rate (Fig. 5l; two-tailed paired $t$-test: $t(64) = 2.60$, $P = 0.012$, $d = 0.39$, 95% CI = [0.01, 0.08]) and memory strength (Fig. 5m; $t(64) = 2.01$, $P = 0.049$, $d = 0.27$, 95% CI = [0.0007, 0.34]) compared to out-region stimuli that had a comparable memorability score (within ±2 SD of the mean of the in-region stimuli). Together, we have shown that feature neurons are associated with encoding image memorability (see Supplementary Information for

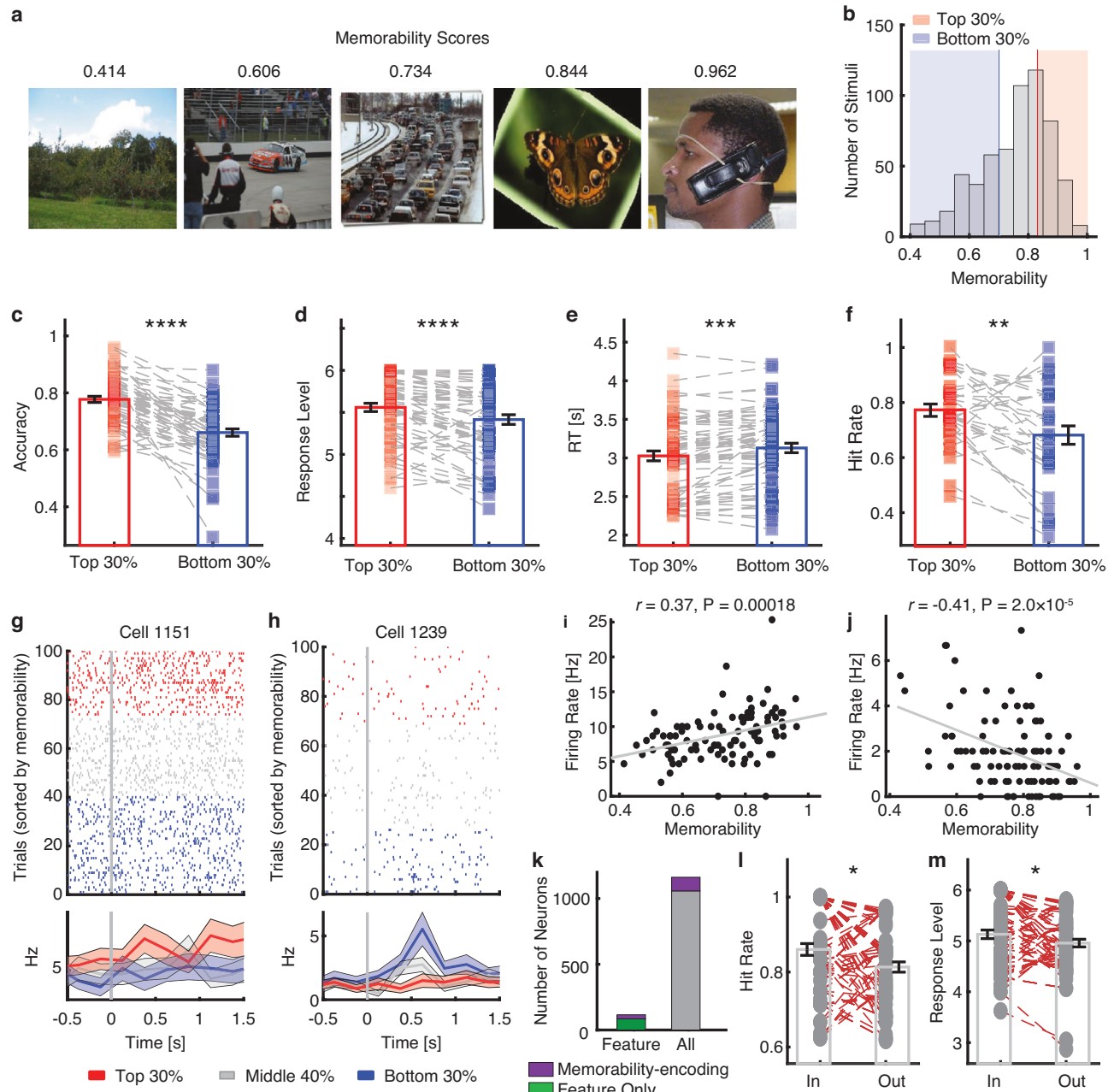

**Fig. 5 | Association between feature neurons and image memorability.**
**a** Example images with memorability scores shown on top. Images are from Faraut et al[38]. Data from: Dataset of human medial temporal lobe single-neuron activity during declarative memory encoding and recognition [Dataset]. Dryad. https://doi.org/10.5061/dryad.46st5. **b** Distribution of memorability scores across all stimuli used in the phased recognition memory task. **c**–**f** Image memorability predicted memory performance. **c** Accuracy (i.e., the proportion of correctly judged trials, including both hits and correct rejections). **d** Response level. **e** Reaction time (RT), relative to stimulus onset. **f** Hit rate (i.e., the proportion of trials correctly recognizing "old" stimuli). Each square represents a session, and error bars denote ±SEM across sessions. Asterisks indicate a significant difference between high-memorability (top 30%) vs. low-memorability (bottom 30%) stimuli using a two-tailed paired *t*-test. \*\**P* < 0.01, \*\*\**P* < 0.001, and \*\*\*\**P* < 0.0001. **g**, **h** Example neurons that differentiated levels of image memorability. Trials are aligned to the stimulus onset.

Shaded area denotes ±SEM across trials. **i**, **j** The firing rate of these example neurons was significantly correlated with image memorability scores (Pearson correlation). Each dot represents an image, and the gray line shows the linear fit. **g**, **i** Cell 1151. **h**, **j** Cell 1239. **k** The number of memorability-encoding neurons among feature neurons (left) and in the whole population (right). Purple: the number of memorability-encoding neurons ($n = 104$ for the whole population and $n = 31$ for feature neurons). Green: the number of non-memorability-encoding feature neurons ($n = 85$). Gray: the number of non-memorability-encoding neurons ($n = 1058$). **l**, **m** Comparison of recognition performance for stimuli that fell within vs. outside feature neurons' tuning regions identified during the learning phase ($n = 65$). Only out-region stimuli with a comparable memorability score as in-region stimuli (i.e., within ±2 SD of the mean of the in-region stimuli) were analyzed. Asterisks indicate a significant difference between in-region vs. out-region stimuli using a two-tailed paired *t*-test. \**P* < 0.05. Legend conventions as in Fig. 4. Source data are provided as a Source Data file.

further comparison between feature neurons and memory-encoding neurons).

We next sought to address whether the above results generalized to tasks without an explicit memory component. To answer this question, we calculated the memorability scores for the ImageNet and COCO stimuli. For the ImageNet stimuli, we found that feature neurons had a significantly higher proportion of neurons that discriminated levels of image memorability (25/89, 28.09%, binomial $P = 1.38 \times 10^{-13}$),

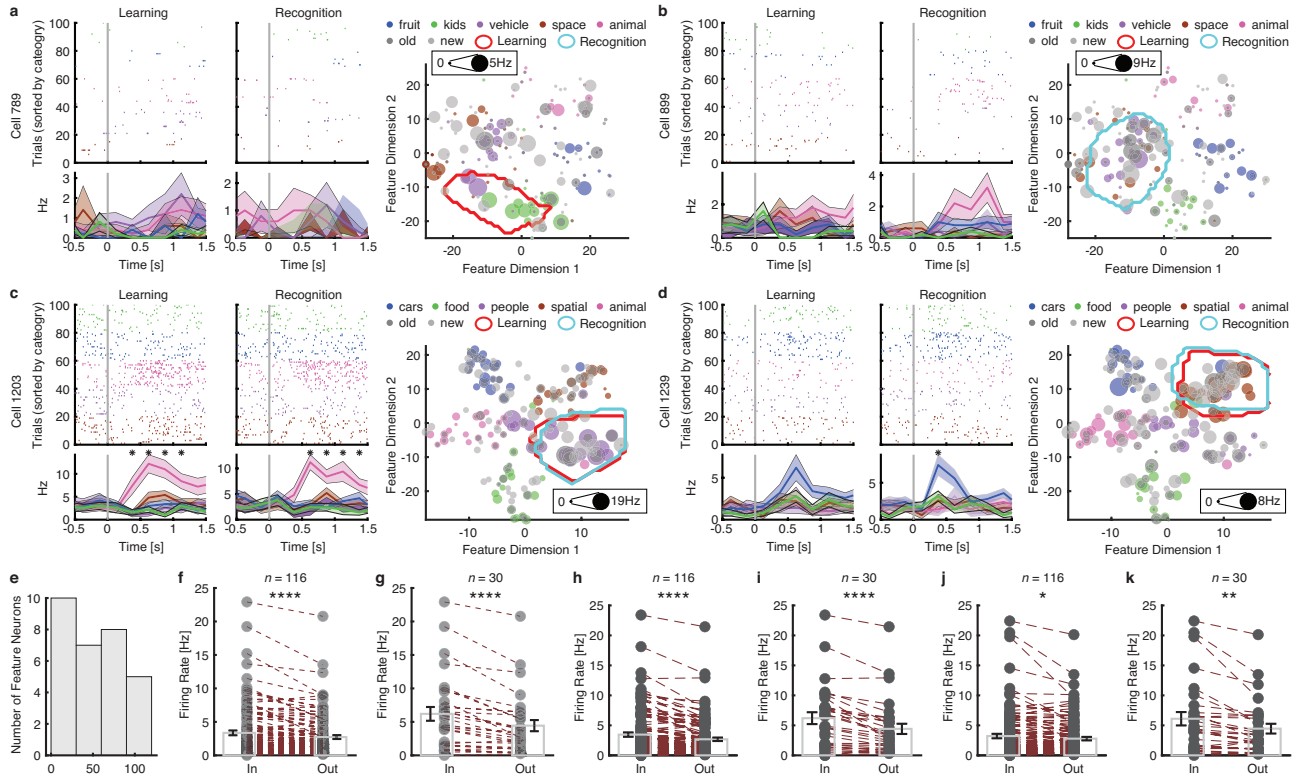

**Fig. 6 | Memory modulation of region-based feature coding. a** An example neuron showing region-based feature coding in the learning phase only. **b** An example neuron showing region-based feature coding in the recognition phase only. **c**, **d** Example neurons showing similar region-based feature coding in both learning (red outlines) and recognition (cyan outlines) phases. Legend conventions as in Fig. 1. In the PSTH, trials are aligned to the stimulus onset. Shaded area denotes ±SEM across trials. Asterisk indicates a significant difference between categories in that bin (*P* < 0.05, one-way ANOVA, after Bonferroni correction across 10 time bins; bin size = 250 ms; no significant bin in **a** and **b**). **e** The percentage of region overlap between learning and recognition for feature neurons identified in the learning phase. **f**, **k** Neural response between in-region vs. out-region stimuli in the recognition phase. Each circle represents a feature neuron, and error bars denote ±SEM across feature neurons. Asterisks indicate a significant difference between in-region vs. out-region stimuli across feature neurons using a two-tailed paired *t*-test. *P < 0.05, **P < 0.01, and ****P < 0.0001. **f**, **g** All stimuli. **h**, **i** "Old" stimuli only. **j**, **k** "New" stimuli only. **f**, **h**, **j** All feature neurons (*n* = 116). **g**, **i**, **k** Feature neurons jointly selective in both learning and recognition phases (*n* = 30). Source data are provided as a Source Data file.

compared to all recorded neurons (77/874, 8.81%, binomial $P = 9.00 \times 10^{-7}$; $\chi^2$-test between feature and all: $P = 1.79 \times 10^{-8}$). We also found that 43 out of 89 feature neurons had a significantly different memorability score between in-region and out-region stimuli. For the COCO stimuli, again, we found that 43 out of 61 feature neurons had a significantly different memorability score between in-region and out-region stimuli. Therefore, feature neurons still encoded image memorability even when the tasks did not involve explicit memory decisions.

Finally, we replicated our findings with fixation-based analysis in the continuous recognition memory task (Supplementary Information).

Together, our results suggest that, in addition to memory performance, feature neurons with region-based coding are also more likely to encode memorability, both in tasks with and without explicitly reported memory performance.

**Context dependency of region-based feature coding on memory**
We finally investigated how memory modulated region-based feature coding in human MTL neurons. In the phased recognition memory task, we investigated whether feature neurons showed a consistent response across task phases (the same neurons as in Fig. 4). While the majority of feature neurons appeared selective in only 1 task phase (86/116 in the learning phase and 110/140 in the recognition phase; see Fig. 6a, b for examples), a significant percentage of neurons (*n* = 30; $\chi^2$-test against the whole population: *P* = 0.002) exhibited region-based

feature coding across task phases. Among these feature neurons, 16 exhibited invariant coding across task phases (i.e., quantified as having more than 50% region overlap; see Fig. 6c, d for examples; see Fig. 6e for the distribution of the percentage of region overlap; see also Supplementary Fig. 3g–j for positive control analyses demonstrating consistency within tasks), while the remaining 14 exhibited more flexible coding of regions (Fig. 6e; see Supplementary Fig. 2e, j for a breakdown of amygdala and hippocampal neurons). Critically, we investigated whether the tuning regions identified in the learning phase could predict responses to stimuli in the recognition phase. Indeed, both the entire population of feature neurons (Fig. 6f; $t(115) = 5.15$, $P = 1.09 \times 10^{-6}$, $d = 0.18$, 95% CI = [0.39, 0.87]) and the feature neurons jointly selective across task phases (Fig. 6g; $t(29) = 5.20$, $P = 1.46 \times 10^{-5}$, $d = 0.34$, 95% CI = [1.07, 2.45]) elicited a higher response for in-region stimuli compared to out-region stimuli in the recognition phase. Notably, this was the case not only for "old" stimuli present in both task phases (Fig. 6h, i) but also for new stimuli presented in the recognition phase that was novel to the patients (Fig. 6j, k). Together, our results suggest that memory modulates region-based feature coding, and feature neurons may be differentially involved in learning and recognition. However, there is also consistency in tuning regions between the learning and recognition phases, which retains predictability in neural response. The balance between invariant and flexible encoding by feature neurons may facilitate efficient memory encoding and retrieval (see "Discussion" section; note that neurons were sorted across task phases and exhibited consistent responses throughout).

## Discussion

In this study, we conducted a comprehensive analysis of 3173 neurons across four single-unit datasets, each containing independent object and natural scene image sets. We found that neurons in the human amygdala and hippocampus exhibited a region-based feature code for general objects, explaining visual category selectivity in these brain regions. Importantly, our findings were consistently replicated across all datasets, confirming their robustness; and we also validated our results with fixation-based analyses for objects from large natural scene stimuli. Furthermore, we explored the link between region-based feature coding and memory and found that objects encoded in feature neurons' tuning regions were better retained in memory. Additionally, we discovered that feature neurons played a role in encoding image memorability. Lastly, we showed how memory modulated neural object coding during learning and recognition. Together, our study provides new insights into the neural mechanisms underlying object coding in the human amygdala and hippocampus.

### Region-based feature coding of visual objects

The amygdala and hippocampus are located downstream from the higher visual cortex, where feature-based coding for faces[25,29] and objects[20] are observed. However, no feature-based encoding of objects has been reported so far in the human amygdala and hippocampus. Instead, only exemplar-based coding has been demonstrated[15,16]. This suggests a fundamental difference in how neural representations are structured within the MTL compared to the higher visual cortex. A key question is how visual feature information (i.e., feature-based coding) is transformed into object category representations (i.e., exemplar-based coding) for memory. Together with our prior study on faces[31], we describe a type of neuron that encodes an intermediate type of representation between these two formats. Specifically, these previously undiscovered neurons encode a region within the high-level feature space, making them responsive to all stimuli falling into this region, thereby connecting the two distinct coding mechanisms. Therefore, region-based feature coding may serve as the basis for semantic representations (typically organized by visual category) in the MTL, which in turn are the basis for declarative memory[30].

### Region-based feature coding and category selectivity

Population neuronal activity patterns in the MTL are governed by high levels of semantic abstraction, enabling the efficient distillation of sensory experiences into sparse representations and the generalization of knowledge[12]. In this study, we compared region-based feature coding and category selectivity, demonstrating that region coding can serve as a more comprehensive mechanism that explains category selectivity. Specifically, region coding does not presuppose any categorical membership of individual images, as long as they share similar visual features (e.g., Figs. 1b, 2l, m, o, 4d). In particular, some pictures of the same object were not clustered in the feature space, accounting for the misclassification of objects. However, even in cases of misclassification, their responses followed the features (e.g., Fig. 2o). Therefore, feature encoding did not depend on categories but solely on features, and this was the case across experiments. Furthermore, the COCO dataset had fewer categories but contained 50 images per category. This abundance of data per category offered a better opportunity to investigate within-category encoding and to distinguish between feature coding and category coding. In particular, when not all objects within a category were encoded, feature coding was favored over category coding (Figs. 2n and 3a). Furthermore, unlike previous findings that showed category-selective neurons to be orthogonal to memory neurons (i.e., they are insensitive to whether a stimulus is novel or familiar or whether a stimulus is retrieved with high or low confidence)[10], using the same data, we observed that feature neurons predicted memory performance (Fig. 4), indicating their distinct involvement in

memory processes compared to category-selective neurons (details shown in ref. 10).

### Region-based feature coding and memory

In this study, we demonstrated that region-coding neurons were linked to aspects of declarative memory. Specifically, we found that stimuli within these neurons' tuning regions were not only better remembered but also associated with greater memory strength (Fig. 4), suggesting that stimuli encoded through region-based feature coding may be preferentially stored in memory. Furthermore, region-coding neurons were more likely to encode memorability, both in tasks with and without explicitly reported memory performance (Fig. 5), indicating that region coding may effectively connect perceptual features to memory. While familiar or personally relevant stimuli are preferentially encoded by MTL neurons[43], and we have shown that MTL neurons encode familiarity[44], it is worth noting that across experiments, we used unfamiliar and non-personally relevant pictures as stimuli to minimize the influence of stimulus familiarity (see also ref. 38 for details). Furthermore, it has been shown in the phased recognition memory task that category selectivity is orthogonal to memory[10], and we have shown that region-based feature coding is independent of face familiarity[31]. Therefore, better recognition of stimuli encoded by feature neurons could not be simply driven by stimulus familiarity.

Notably, feature neurons strike a balance between invariant and flexible encoding during both the learning and recognition phases (Fig. 6). On the one hand, a subset of feature neurons encoded tuning regions consistently and without change, demonstrating invariant encoding. Invariant encoding ensures that well-established memories remain stable and retrievable, which is essential for long-term memory. On the other hand, another subset of feature neurons encoded tuning regions in only one task phase or encoded separate tuning regions, demonstrating flexible encoding. Flexible encoding allows us to adapt to new situations and acquire new memories. It ensures that we can integrate new information into our existing knowledge base. Previous research has demonstrated that individual neurons in the human MTL can flexibly shift representations across spatial and memory tasks[45] (see ref. 46 for a review). In our previous studies, we have also shown task modulation of MTL neuron activity in social perception tasks[47] and flexible BOLD-fMRI responses in neural face representations[29] and approachability[48] in the human amygdala. Our results also indicate that feature neurons may constitute functionally different subgroups and have diverse roles in memory processes, including encoding, retrieval, and saliency. Future studies are needed to explore the full extent of their involvement in memory.

The advantages of our DNN-based approach to studying neural object coding as well as the limitations of the present study are discussed in Supplementary Information.

## Methods
### Participants

We recruited 11 neurosurgical patients from the West Virginia University (WVU), 8 patients from the Washington University in St. Louis (WUSTL), 5 patients from the Cedars-Sinai Medical Center (CSMC), and 1 patient from the University of Utah. All participants provided written informed consent using procedures approved by the Institutional Review Board.

Six patients from WVU, one patient from Utah, and eight patients from WUSTL performed the one-back task with the ImageNet stimuli, yielding 874 neurons with an overall mean firing rate greater than 0.15 Hz (Supplementary Table 1). Five patients from WVU and eight patients from WUSTL performed the one-back task with the Microsoft COCO stimuli, totaling 453 neurons. Additionally, five patients from WVU (separate from those who performed the above one-back tasks) and five patients from CSMC participated in the continuous recognition memory task, yielding 684 neurons. Furthermore, we analyzed an

existing dataset involving 40 patients in a phased recognition memory task[38], which contributed 1162 neurons. Overall, we analyzed a total of 3173 neurons with an overall mean firing rate greater than 0.15 Hz. Different sessions/tasks were recorded on different days and spikes were sorted separately for each session/task, except for the analysis shown in Fig. 3c, d where the ImageNet and Microsoft COCO stimuli were recorded in the same session.

### Experimental procedure and stimuli: one-back task and ImageNet stimuli

We used a one-back task for the ImageNet stimuli. In each trial, a single object image was presented at the center of the screen for a fixed duration of 1 s, with a uniformly jittered inter-stimulus-interval (ISI) of 0.5–0.75 s. Each image subtended a visual angle of approximately 10°. Patients pressed a button if the present image was *identical* to the immediately previous image. 10% of trials were one-back repetitions. Each image was shown once unless repeated in one-back trials, and we excluded responses from one-back trials to have an equal number of responses for each image. This task kept patients attending to the images but avoided potential biases from focusing on a particular image feature[49].

We selected 50 categories of objects with 10 images for each object category from the ImageNet dataset[50]. The object categories included arachnid, battery, bark, beverage, board, bread, brier, building, car, cat, collection, crustacean, dainty, dog, electrical device, electronic device, equipment, fare, fern, fish, flower, frog, fruit, fungus, furniture, game bird, gymnast, herb, hole, insect, light, man clothing, moped, musical instrument, needlework, nest, plate, reptile, ridge, rock, rodent, star, sugar maple, support, tool, utensil, vegetable, vessel, weapon, and young mammal.

### Experimental procedure and stimuli: one-back task and Microsoft COCO stimuli

We used the same one-back task for the Microsoft COCO stimuli. We selected 10 categories of objects with 50 images for each object category from the Microsoft COCO dataset[51]. The object categories included airplane, apple, bear, bird, car, chair, dog, elephant, person, and zebra.

### Experimental procedure and stimuli: phased recognition memory task

There were three task variations, all of which were the same except for the images displayed (213, 156, and 225 images, respectively). In each stimulus set, there were images selected from five distinct visual categories, with an equal number of instances from each category. The experiment was divided into two phases: a learning phase and a recognition phase. In the learning phase, participants viewed 100 unique novel images, each displayed only once for either 1 or 2 s. They were instructed to pay close attention to these images for a subsequent memory test, focusing on forming explicit memories. As a control measure, participants indicated after each learning trial whether the image contained an animal or not. In the recognition phase, a random subset of 50 of these previously shown images, now considered "old", was randomly mixed with a new set of 50 novel images. Following the display of each image, participants were asked to determine whether they had seen that exact image before ("old") or not ("new"), and to express their confidence using a 1 to 6 confidence scale: 1 = new, very sure; 2 = new, sure; 3 = new, guess; 4 = old, guess; 5 = old, sure; 6 = old, very sure. Participants responded by pressing buttons on an external response box (RB-740, Cedrus Inc.). Responding was only possible after the question screen appeared, and there was no time constraint for response. The question screen remained on display until the participant provided an answer, without any timeout.

We included all trials for the neural analyses. To compare behavior, we excluded trials with outlier response times (those longer than 5 s or falling outside the average plus 3 standard deviations). On average, 94.75% of trials were retained in the learning phase, and 88.07% were retained in the recognition phase.

### Experimental procedure and stimuli: continuous recognition memory task

We employed a continuous recognition memory task with natural scene images (Supplementary Fig. 4a). We had both learning sessions and recognition sessions, but both sessions had the same task (see below). In the learning session, 100 unique images were selected, and 50 of these images were repeated. Therefore, the learning session had 150 trials in total. In the recognition session, another 100 images were selected, comprising the "new" stimuli. All 100 images from the learning session were shown again, comprising the "old" stimuli. Patients viewed each image for 3 s. After each image, patients were asked "Have you seen this image before?" and they were required to respond as soon as possible on a six-point scale (no sure, no less sure, no unsure, yes unsure, yes less sure, yes sure). Recognition sessions followed the corresponding learning sessions, either after 30 min or the next day. Some patients underwent two separate learning and recognition sessions, and we used a different and non-overlapping subset of images for different learning and recognition sessions. This procedure has been demonstrated to be very effective in studying memory and has been used extensively in previous human single-neuron studies[7,10,52].

We used natural scene images from the OSIE dataset[53]. This dataset has been characterized and described in detail previously[53,54]. Briefly, the dataset contains 700 images, which have been quantified according to three pixel-level attributes (color, intensity, and orientation), five object-level attributes (size, complexity, convexity, solidity, and eccentricity), and twelve semantic attributes (face, emotion, touched, gazed, motion, sound, smell, taste, touch, text, watchability, and operability) annotated on a total of 5551 segmented objects. Since there are a large number and variety of objects in natural scenes, to make the ground truth data least dependent on subjective judgments, we followed several guidelines for the segmentation, as described in ref. 53. Similar hand-labeled stimuli[55] have demonstrated advantages in understanding the saliency contributions from semantic features. Images contain multiple dominant objects in a scene. The twelve semantic attributes fall into four categories: (i) directly relating to humans (i.e., face, emotion, touched, gazed); (ii) objects with implied motion in the image; (iii) relating to other (non-visual) senses of humans (i.e., sound, smell, taste, touch); and (iv) designed to attract attention or for interaction with humans (i.e., text, watchability, operability). We used the contents of fixations on faces and objects as the input to construct the feature space.

### Feature extraction and construction of feature space

We employed two well-known deep neural networks (DNNs), AlexNet[56] and ResNet[57], to extract features for each image. Following the same procedure[31], fine-tuning of the top layer of each DNN was performed to confirm that the pre-trained model was able to discriminate the objects and ensure that the pre-trained model was suitable as a feature extractor. We also used fine-tuning accuracy to determine the most suitable model for feature extraction.

We subsequently applied a t-distributed stochastic neighbor embedding (t-SNE) method to convert high-dimensional features into a two-dimensional feature space. t-SNE is a variation of stochastic neighbor embedding (SNE)[58], a commonly used method for multiple-class high-dimensional data visualization[59]. We applied t-SNE for each layer, with the cost function parameter (Perp) of t-SNE, representing the perplexity of the conditional probability distribution induced by a Gaussian kernel, set individually for each layer. Because a sparse distribution of objects could lead to a larger tuning region, we adjusted the distribution of objects using the t-SNE perplexity parameter so that

the objects were distributed approximately homogeneously. A robustness analysis of the perplexity parameter and Gaussian kernel was conducted as in our previous study[31] to ensure that our results were robust to these parameters.

It is worth noting that neither feature extraction nor construction of feature spaces utilized any information from neurons. Therefore, the clustering of neurally encoded categories in feature spaces was not by construction.

For the phased recognition memory task, features for each stimulus in the three task variants were extracted using AlexNet. The model performed well in predicting the categories for the three sets of stimuli, achieving cross-validation accuracies of $95.84\% \pm 2.06\%$ for Variant 1, $95.52\% \pm 2.52\%$ for Variant 2, and $98.58\% \pm 1.16\%$ for Variant 3. A feature space was constructed separately for each task variant. For each neuron, only the displayed stimuli ($n = 100$) were included in the statistics. The size of the space might change slightly because only half of the stimuli remained the same across the learning phase and recognition phase. To facilitate direct comparisons and cross-validation between the two phases, we consistently included the four corner coordinates in the space constructed with all possible stimuli in each variant.

### Electrophysiology

We recorded using implanted depth electrodes in the amygdala and hippocampus from patients with pharmacologically intractable epilepsy. Target locations in the amygdala and hippocampus were determined by the neurosurgeon based solely on clinical need and verified using post-implantation CT. At each site, we recorded from eight 40 μm microwires inserted into a clinical electrode. Bipolar wide-band recordings (0.1–9000 Hz), using one of the eight microwires as a reference, were sampled at 32 kHz and stored continuously for offline analysis with a Neuralynx or Blackrock system[60]. The raw signal was filtered with a zero-phase lag 300–3000 Hz bandpass filter and spikes were sorted using a semi-automatic template matching algorithm[61]. Units were carefully isolated and recording and spike sorting quality were assessed quantitatively (Supplementary Fig. 1).

Consistent with our previous studies[11,62–65], only single units with an average firing rate of at least 0.15 Hz throughout the entire task were considered. Trials were aligned to stimulus onset and fixations were aligned to fixation onset. For trial-based analysis, we used the mean firing rate in a time window of 250–1250 ms after stimulus onset as the response to each object. For fixation-based analysis, we used the mean firing rate in a time window of 0–300 ms after fixation onset as the response to each fixation. The rationale behind using different response windows for trial-based and fixation-based analyses lies in the nature of the neural responses being studied. Trial-based analysis focuses on the neural activity in response to the entire presented stimuli, while fixation-based analysis captures transient neural responses associated with fixational eye movements and fixated stimuli. Although the response windows did not overlap significantly, they may capture distinct aspects of neural processing relevant to neural object coding. Importantly, we observed region-based feature coding for both trial-based and fixation-based analyses.

### Eye tracking

Patients were recorded with a remote non-invasive infrared Eyelink 1000 system (SR Research, Canada). One of the eyes was tracked at 500 Hz. The eye tracker was calibrated with the built-in 9-point grid method at the beginning of each block. Fixation extraction was carried out using software supplied with the Eyelink eye tracking system. Saccade detection required a deflection of greater than 0.1°, with a minimum velocity of 30°/s and a minimum acceleration of 8000°/s², maintained for at least 4 ms. Fixations were defined as the complement of a saccade, i.e., periods without saccades. Analysis of the eye movement record was carried out offline after completion of the experiments. Each fixation was treated individually, and multiple consecutive fixations that fell on the same object were counted as discrete samples.

### Selection of region-coding feature neurons

To select region-coding feature neurons, we first estimated a continuous spike density map in the feature space by smoothing the discrete firing rate map using a 2D Gaussian kernel. The kernel size was proportional to the number of clusters (where images from the same category would form a cluster) within each feature space, feature space dimension, and an empirical scaling factor (sq) estimated for each feature space (ImageNet: sq = 0.021; COCO: sq = 0.05; phased recognition memory: sq = 0.11; continuous recognition memory: sq = 0.03). We then estimated the statistical significance for each pixel ($100 \times 100$ grid of the feature space) using permutation testing: in each of the 1000 runs, we randomly shuffled the labels of the objects. We calculated the p-value for each pixel by comparing the observed spike density value to those from the null distribution derived from permutation. We applied a mask to exclude pixels from the edges and corners of the spike density map where there were no objects because these regions were susceptible to false positives given our procedure. We lastly selected the region with significant pixels (permutation $P < 0.01$, cluster size > 2.5% of the pixels within the mask). If a neuron had a region with significant pixels, the neuron was defined as a *feature neuron* and demonstrated *region-based feature coding*. We selected feature neurons for each individual DNN layer. Our previous study has shown that this procedure is effective in identifying feature neurons[31].

We confirmed that our results were robust to different feature metrics. Similar results were derived if we constructed a three-dimensional feature space or used different perplexity parameters for t-SNE (balance between local and global aspects of the data) or kernel/cluster size parameters to detect a tuning region (balance between sensitivity and specificity of detecting a tuning region). Similar results were also derived if we constructed the feature space using other common methods, such as uniform manifold approximation and projection (UMAP) or principal component analysis (PCA). Therefore, our findings were robust to the construction of the feature space. Note that the 2D feature space was used for subsequent analyses because it offered more straightforward interpretation and visualization of results compared to the full DNN features, which could be more complex and difficult to interpret.

Furthermore, we could also replicate our findings using full DNN features, where the Euclidian distance between encoded categories was significantly smaller than that of non-encoded categories. Specifically, feature distance was calculated between categories using the average DNN full feature of objects from each category (Fig. 1d). Feature distance was then normalized by the maximum DNN full feature distance. We first averaged the feature distance of all selective-selective (S-S) category pairs and all selective-non-selective (S-NS) category pairs for each feature MC neuron, and then we compared the feature distance between S-S vs. S-NS category pairs across neurons. We confirmed that the feature distance was significantly shorter for S-S category pairs, suggesting that feature MC neurons encoded categories that were clustered in these layers.

### Selection of axis-coding neurons

To identify axis-coding neurons, i.e., neurons that encoded a linear combination of visual features, we employed a partial least squares (PLS) regression with DNN feature maps. This PLS method has been shown to be effective in studying the neural response to DNN features[28,66]. We used 10 components for each layer (explaining at least 80% of the variance; we selected the number of components with a 10-fold cross-validation to minimize the prediction error) and a permutation test with 1000 runs to determine whether a neuron encoded a significant axis-coding model. In each run, we randomly shuffled the

object labels and used 50% of the objects as the training dataset. We used the training dataset to construct a model (i.e., deriving regression coefficients), predicted responses using this model for each object in the remaining 50% of the objects (i.e., test dataset), and computed the Pearson correlation between the predicted and actual response in the test dataset. The distribution of correlation coefficients computed *with* shuffling (i.e., null distribution) was eventually compared to the one *without* shuffling (i.e., observed response). If the correlation coefficient of the observed response was greater than 95% of the correlation coefficients from the null distribution, this axis-coding model was considered *significant*. This procedure has been shown to be very effective in selecting units with significant face models[26]. The correlation coefficient could also indicate the model's predictability and thus be compared between different neurons.

### Selection of category-selective neurons

To select category-selective neurons, we first used a one-way ANOVA to identify neurons with a significantly unequal response to different object categories. We next imposed an additional criterion to identify the *selected categories*: the neural response to such a category was required to be at least 1.5 standard deviations (SD) above the mean neural response during baseline (−500 to 0 ms relative to stimulus onset) to all categories. We refer to the neurons that encoded a single object category as single-category (SC) neurons and we refer to the neurons that encoded multiple categories as multiple-category (MC) neurons. Our previous study has shown that this procedure is effective in identifying category-selective neurons[31].

### Binomial test

We used the binomial test to determine whether the observed proportion of selected neurons differed significantly from a given chance level. Specifically, the null hypothesis assumed that the selection of neurons was due to chance (5%), while the alternative hypothesis suggested that the selection was greater than chance. The binomial test calculates the probability of observing at least $k$ selected neurons out of a total of $n$ neurons, given the probability $p$, using the cumulative distribution function (CDF) of the binomial distribution. The $p$-value represents the probability of obtaining the observed number of selected neurons (or more) under the null hypothesis:

$$\text{binomial test } p\text{-value} = P(X \geq k) = 1 - \sum_{i=0}^{k-1} \binom{n}{i} p^i (1-p)^{n-i}$$

This approach allows us to statistically assess whether the observed number of selected neurons is likely due to a real effect rather than random chance.

### Category selectivity index

To assess each neuron's selectivity to different object categories, we defined a category selectivity index as the $d'$ between the most-preferred and least-preferred object categories:

$$\text{Category Selectivity Index} = \frac{\mu_{\text{best}} - \mu_{\text{least}}}{\sqrt{\frac{1}{2}\left(\sigma^2_{\text{best}} + \sigma^2_{\text{least}}\right)}}$$

where $\mu_{\text{best}}$ and $\mu_{\text{worst}}$ denote the mean firing rate for the most-preferred and least-preferred categories, respectively, and $\sigma^2_{\text{best}}$ and $\sigma^2_{\text{worst}}$ denote the variance of firing rate for the most-preferred and least-preferred categories, respectively. A similar index was used in previous studies to assess the level of selectivity to different faces[31,49]. It is worth noting that the category selectivity index was not used to select category-selective neurons or estimate the number of neurons that were category-selective. Instead, the category selectivity index was used to quantify the degree of category selectivity for the category-selective and non-category-selective neurons that had already been selected.

### Response ratio

Response ratio was calculated for each object category by first dividing by the response of the most-preferred category and then ranking the categories from the most preferred to the least preferred. The response ratio of the most-preferred category is thus 1. We compared the response ratio for each ordered category between SC/MC vs. non-category-selective neurons using a two-tailed two-sample $t$-test (corrected for multiple comparisons using false discovery rate [FDR][67]). A steeper change from the best to the worst category indicates a stronger category selectivity.

### Depth of selectivity (DOS) index

To summarize the response of category-selective neurons, we quantified the depth of selectivity (DOS) for each neuron: $DOS = \frac{n - (\sum_{j=1}^{n} r_j)/r_{\max}}{n-1}$, where $n$ is the number of categories ($n = 50$), $r_j$ is the mean firing rate to category $j$, and $r_{max}$ is the maximal mean firing rate across all categories. DOS varies from 0 to 1, with 0 indicating an equal response to all categories and 1 exclusive response to one category, but not to any of the other categories. Thus, a DOS value of 1 is equal to the maximal sparseness of category coding. The DOS index has been used in many prior studies investigating visual selectivity[11,68,69].

### Population decoding of object categories

We pooled all recorded neurons into a large pseudo-population. Firing rates were $z$-scored individually for each neuron to give equal weight to each unit regardless of firing rate. We used a maximal correlation coefficient classifier (MCC) as implemented in the MATLAB neural decoding toolbox (NDT)[70]. The MCC estimates a mean template for each class $i$ and assigns the class for a test trial. We used 8-fold cross-validation, i.e., all trials were randomly partitioned into 8 equal-sized subsamples, of which 7 subsamples were used as the training data and the remaining single subsample was retained as the validation data for assessing the accuracy of the model, and this process was repeated 8 times, with each of the 8 subsamples used exactly once as the validation data. We then repeated the cross-validation procedure 50 times for different random train/test splits. Statistical significance of the decoding performance for each group of neurons against chance was estimated by calculating the percentage of bootstrap runs (50 in total) that had an accuracy below chance (i.e., 2% when decoding all object categories). Statistical significance for comparing between groups of neurons was estimated by calculating the percentage of bootstrap runs (50 in total) that one group of neurons had a greater accuracy than the other. Spikes were counted in bins of 500 ms size and advanced by a step size of 50 ms. The first bin started at −500 ms relative to trial onset (bin center was thus 250 ms before trial onset), and we tested 31 consecutive bins (the last bin was thus from 1000 ms to 1500 ms after trial onset). For each bin, a different classifier was trained/tested. For both tests, we used FDR[67] to correct for multiple comparisons across time points. The same decoding approach was used in our prior studies[10,71] and has been shown to be very effective in the study of neural population activity.

### Image memorability

We used a pre-trained DNN model, ResMem[40], designed to predict the memorability of images, to calculate the memorability score for each image. The memorability score from ResMem represents how likely an image is to be remembered by people after a single view. The score ranges from 0 to 1, with scores closer to 1 indicating that the image is highly memorable, meaning most people who see it are likely to remember it, and scores closer to 0 suggesting that the image is less memorable, meaning people are less likely to recall it after seeing it. This score is derived from patterns learned by the ResMem model, which was trained on large datasets of images with known memorability ratings. The model analyzes the visual features of an image and predicts

how effectively it will be stored in and retrieved from human memory. For each dataset, we compared stimuli with the top 30% memorability scores (high-memorability group) to stimuli with the bottom 30% memorability scores (low-memorability group) across sessions. To ensure a reliable comparison, we only included sessions with more than 10 trials in both the high-memorability and low-memorability groups (e.g., 62 out of 64 sessions from the phased recognition memory task).

## Reporting summary
Further information on research design is available in the Nature Portfolio Reporting Summary linked to this article.

## Data availability
All data that support the findings of this study are publicly available on OSF. Source data are provided with this paper.

## Code availability
The source code for this study is publicly available on OSF.

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

## Acknowledgements

We thank all patients for their participation and staff from WVU Ruby Memorial Hospital, Barnes-Jewish Hospital, Cedars-Sinai Medical Center, and University of Utah Hospital for support with patient testing. This research was supported by the AFOSR (FA9550-21-1-0088 [S.W.]), NSF (BCS-1945230 [S.W., X.L.], IIS-2114644 [X.L., S.W.]), NIH (K99EY036650 [R.C.], R01MH129426 [S.W., X.L.], R01MH120194 [J.T.W.], R01EB026439 [P.B.], U24NS109103 [P.B.], U01NS108916 [P.B.], U01NS128612 [P.B.], R21NS128307 [P.B.], P41EB018783 [P.B.]), McDonnell Center for Systems Neuroscience [R.C.], Fondazione Neurone [P.B.], and Dana Foundation [S.W.]. The funders had no role in study design, data collection and analysis, decision to publish, or preparation of the manuscript.

## Author contributions

R.C., X.L., U.R., and S.W. designed research. R.C., P.B., K.L.W., C.I., and E.H.S. performed experiments. A.N.M., N.J.B., and J.T.W. performed the surgery. R.C., P.N.C., X.L., and S.W. analyzed data. R.C., U.R., J.T.W., and S.W. wrote the paper. All authors discussed the results and contributed toward the manuscript.

## Competing interests

The authors declare no competing interests.
