## [Transparent Peer Review file · Nature Communications]

A neuronal code for object representation and memory in the human amygdala and hippocampus

Corresponding Author: Dr Shuo Wang

Version 0:

Reviewer comments:

Reviewer #1

(Remarks to the Author)

This study examined the neural code of objects in the human medial temporal lobe. More than one thousand neurons were recorded in the amygdala and hippocampus of neurosurgical patients while they viewed natural object images. Using a DNN model to construct an object space, analyses revealed a region-based encoding scheme, which differs from the axis-based coding in the macaque inferotemporal cortex. The authors also claim that this region-based coding predicts the image memorability and is modulated by memory. Overall, this is an interesting study that elucidates the code for object representation and memory based on rare single-unit data from the human brain and sophisticated analyses using machine learning techniques. However, I have several concerns about the interpretation of the results that need to be addressed:

1) Category selectivity vs. feature selectivity. One of the main conclusions of this study is that visual features explain the responses of MTL neurons better than object category. Here, the “category” seems to be defined by the original class label in the image database. One problem is that many natural images contain more than one type of object, but only one was labeled in each image. While the “apple” image in Figure 2o is used to argue that the neuron doesn’t encode the category “apple”, it is clear that the leftmost example image contains not only apples but also furniture, which also appears in other preferred images, weakening the claim that the neuron doesn’t encode object category.

2) Definition of memorability. Figure 5 shows a complex relationship between the memorability of an image, the relationship between the image and the neuron’s tuning region, and task performance. The authors defined memorability using a pre-trained DNN. Both DNN-estimated memorability and region-based neural preference were positively correlated with performance, but the two indices were negatively correlated (Figure 5L). This suggests a dissociation between memorability and performance, as otherwise they should have a similar relationship with the region-based neural preference. Since the task used here is basically a recognition memory task, and what it measures is the memorability of the image, its dissociation with the DNN estimate simply indicates that the estimated memorability is inaccurate. I think the DNN estimated scores here are quite confusing, and the authors could simply remove them. Otherwise, please clarify why the estimated scores differ from the subjects’ performance.

3) Memory modulation needs to be tested statistically. The title of Figure 6 is “Memory modulation of region-based feature coding”, but it seems that only the consistency of the preferred region between two phases has been statistically tested, not its modulation by task. The main evidence may be the 14 neurons with flexible coding of regions, but it’s unclear whether the difference in estimated regions is simply due to noise in the data. This could be addressed by splitting the data into an earlier and a later half, and comparing the two estimated tuning regions.

Minor:

1) Page 5: “Feature Dimension 1 represented changes in object animacy”. The description seems inaccurate, as flowers appear on the positive side of dimension 1, along with animal images.

2) Page 5: “We then projected the response of a given neuron to each object onto the feature space by multiplying the response magnitude to each object to its corresponding location in the feature space, resulting in a response-weighted 2D feature map”. Since the objects occupy discrete locations in the feature space, how is a continuous 2D feature map obtained?

3)Page 5: “Moreover, we found that only 3.20% (below-chance) of the amygdala and hippocampal neurons exhibited axis-based feature coding, consistent with our prior report examining neuronal responses to faces”. More details are needed. How are axis-based coding neurons defined?

4)Page 7: “whereas SC neurons showed steeper changes from the most- to the least preferred categories”. It’s not so obvious from the figure alone. I understand that the difference between the top two images was quantified in Figure 2E, but not for other images.

5)Page 7: “In addition, single-trial population decoding revealed that MC neurons differentiated between more stimuli than SC neurons”. It’s unclear whether the number of neurons was matched between two groups of neurons.

6)The example cell in Figure 3D is not very compelling. Where is this cell in Figure 3E? It would be nice to see the plot as in Figure 3E, but comparing COCO stimuli within and outside the region, as a positive control. For example, one could use half of the COCO data to estimate the tuning region, and the other half to compute the mean response within and outside the region.

7)Page 14: “we observed that feature neurons predicted memory performance, indicating their distinct involvement in memory processes compared to category-selective neurons”. Where are the results for category-selective neurons?

Reviewer #2

(Remarks to the Author)

I reviewed this manuscript previously for another journal and my opinion remains largely the same. This manuscript describes results from experiments on visual perception and memory using single-unit recordings of the human hippocampus and amygdala in subjects with epilepsy undergoing invasive EEG. The main goals were to identify how hippocampal/amygdala neurons code for high-level stimulus features and how this is related to memory. A neural network model of visual perception was used to project a large set of visual stimuli into a 2-dimensional feature space, including dimensions for object animacy and object size. About 10% of identified hippocampal/amygdala units responded maximally to a particular range of this 2-dimensional feature space, which is described as a “region-based feature code.” A number of control analyses were performed to validate this coding across different stimulus sets and to test accuracy in predicting neural activity relative to category-selective coding. In a recognition memory experiment, stimuli that fell within the feature tuning region of neurons showing the region-based feature code were better recognized with higher confidence than stimuli that fell outside the tuning region of identified neurons. The authors also investigated “memorability”, which refers to the tendency for some stimuli to be recognized consistently across individuals in memory tests, although the findings were somewhat less clear (feature neurons discriminated stimulus memorability better than other neurons, but, counterintuitively, memorability was lower for stimuli that were in-region for feature-sensitive neurons despite better memory performance for these stimuli). Finally, by comparing feature-sensitivity across encoding versus retrieval phases of the task, the authors found that many neurons changed the region of the feature space to which they were maximally selective, whereas other neurons remained consistent, demonstrating memory modulation.

Although many technical aspects of this report are impressive, including a difficult-to-acquire dataset and a number of sophisticated analyses that blend neural network modeling and neural response modeling, I find it difficult to find any major significance in the results. Although the authors make claims about how the findings fundamentally inform understanding of the perceptual-memory transition in hippocampus/amygdala, I consider the impact for perception as subtle, and the link to memory weak and confusing. The main limitation of the report is that it is primarily descriptive and does not include experimental tests needed to support the major conclusions. That is, the authors look for general relationships between feature coding and memory performance but without including relevant manipulations within the task (or within the brain) to test hypotheses about the nature of these relationships. Instead, the general patterns identified in the data motivate hypotheses about brain-perception-memory relationships, which are presented as conclusions.

1. Region-based feature coding of visual objects and category selectivity: A major conclusion is that the identified region-based feature code can “explain”, or is more fundamental than, the category/exemplar-based coding that had previously been identified in these brain areas. However, this conclusion is not fully warranted, as this pattern was the case for neurons that were sensitive to multiple categories (MC), but not those that were sensitive to single categories (SC). A challenge for disentangling these two coding schemes is that feature space and categories are largely correlated, and without specific attempts to de-correlate them or experimental methods to systematically explore divergence, the findings add nuance to the question of how these neural populations code visual features, without being able to strongly support one model over another.

More fundamentally, however, the claims that the findings identify “how the transformation from a perception-driven representation to the memory-based semantic representation in the MTL is achieved” are misplaced, as the study did not compare coding schemes along the visual hierarchy, and did not offer a definition of “semantic representations” with enough specificity to be tested. Qualitative comparisons of how small subsets of neurons respond to visual features among different experiments investigating different brain regions are insufficient for addressing these important questions.

2. Region-based feature coding and memory: A major conclusion is that "feature neurons" (neurons that show selectivity to a region of feature space) "play a crucial role in memory processing." The evidence for this is that stimuli falling into the feature space that was preferred by the recorded neurons were better recognized than those that were not. I am honestly at a loss to understand what this means about how feature neurons contribute to memory, and I do not find that any clear mechanistic explanation was provided.

One point that is very confusing is that only a small subset of the entire population of neurons were recorded. What is the rationale for there being a relationship between stimuli that happen to fall into the feature space of the small subset of neurons that happened to be recorded and memory performance? One possibility is that the portions of feature space coded by feature neurons are particularly salient or have some other special perceptual or semantic property. This could impact perceptual coding and memory, not necessarily through the same mechanism. The findings regarding memorability are equally puzzling, as the relationship is not what would have been expected based on the relationship between feature coding and memory, and the interpretation is very post hoc. The authors should provide a mechanistic test of the relationship between feature neuron coding and memory as the correlation reported is hard to interpret.

3. For the main analysis showing that amygdala and hippocampal neurons exhibit region-based feature coding, only 10% of neurons out of the total of 1204 recorded demonstrated region-based feature coding, with a similar value in the replication experiment. Is this relatively low value predicted by any formal model or a priori considerations? Is there any alternative hypothesis? If not, how is this to be interpreted, in the context of ~90% of neurons not contributing to this function?

4. The authors make conclusions about region-based feature coding reflecting "semantic representations", but there is no operational definition of this term given with sufficient clarity to evaluate this conclusion versus alternatives. There was no experimental manipulation to test whether the encoding processing was "semantic" versus of some other quality. Receptivity to a specific region of a 2D model of image features does not necessarily imply semantic representations. Moreover, some region-based feature coding neurons were specific to different features in different categories, which is opposite to what one would expect if they reflect semantic information processing, which should tend to be organized by category. In general, there are many alternatives, including that neurons may simply respond to stimuli with a certain range of visual complexity. Firm conclusions about function require specification of alternatives, and some modeling or tests sufficient to compare the preferred interpretation against these alternatives.

Reviewer #3

(Remarks to the Author)

Sure, here's the revised version without subtitles:

The paper presents an intriguing study on electrophysiological responses from the amygdala and hippocampus to thousands of natural images under different tasks. The authors demonstrate that neurons' responses can be described as region-based feature codes, akin to receptive fields in a feature space. Additionally, they found that region-based feature-coded neuron responses better predict image memorability compared to non-region-based feature neurons. While the paper offers a novel perspective on the coding scheme, several details regarding the analysis require clarification. Here are my comments:

The paper states that 10% of neurons exhibit a regional coding scheme. However, it is unclear what coding scheme the remaining 90% of neurons follow. If the majority of neurons (90%) are not region-based coding, what alternative coding schemes do they utilize? Additionally, the binomial test referenced in Figure 1 lacks clarity. What is the null hypothesis of this test? It would be beneficial to include this information in the methods section.

The authors claim that only 3.87% of neurons exhibit the axis model. What criteria are used to accept or reject a neuron as following the axis model or the region-based model? The statement "below chance performance" is ambiguous. What is the chance performance of the axis model, and how is it determined? A detailed description of this process should be included in the methods section.

On page 11, in the first paragraph, it is mentioned that 31/116 (26.72%) neurons are feature neurons. However, the paper records 1162 neurons, not 116, and the proportion calculation appears incorrect. Please verify the numbers and correct the proportion accordingly. Additionally, the definition of "whole" in the context of "whole, 104/1162" requires clarification.

The feature space appears to be determined by the stimulus set used (PCA and t-SNE based on responses to the images). It would strengthen the paper to include analyses demonstrating that the feature space is stable across different image datasets, such as COCO images and ImageNet images.

The abstract and discussion emphasize recording over 3k neurons. However, it is noted that most neurons are not included in the analysis. To avoid potential misleading implications, it may be advisable to adjust the emphasis on the total number of recorded neurons.

Version 1:

Reviewer comments:

Reviewer #1

(Remarks to the Author)

The authors have made a good attempt to address all my concerns. I have no further questions.

Reviewer #2

(Remarks to the Author)

The response arguments are mostly convincing and I appreciate that the authors tempered their claims based on the acknowledged limitations. I will note however that the suggestion that the comparison of remembered to forgotten items in the memory test constitutes an experimental manipulation/condition is misleading, as behavior assigns trials to these conditions, not the experimenters, which is non-random selection. Extraneous factors like attentional lapses, drowsiness, or even symptoms related to the patients' neurological condition or intervention could be driving these differences.

I think the results and main conclusions are now mostly convincing, but am still of the opinion that the findings are very narrow: properties of a very small subset of hippocampal neurons that provide modest correlational evidence in favor of one response model over another, without very thorough consideration of relationship to memory more broadly. I'm skeptical that this will have much impact on memory or visual perception fields.

Reviewer #3

(Remarks to the Author)

I have no further comments.

Reply to comments from Reviewer 1

This study examined the neural code of objects in the human medial temporal lobe. More than one thousand neurons were recorded in the amygdala and hippocampus of neurosurgical patients while they viewed natural object images. Using a DNN model to construct an object space, analyses revealed a region-based encoding scheme, which differs from the axis-based coding in the macaque inferotemporal cortex. The authors also claim that this region-based coding predicts the image memorability and is modulated by memory. Overall, this is an interesting study that elucidates the code for object representation and memory based on rare single-unit data from the human brain and sophisticated analyses using machine learning techniques. However, I have several concerns about the interpretation of the results that need to be addressed:

1)Category selectivity vs. feature selectivity. One of the main conclusions of this study is that visual features explain the responses of MTL neurons better than object category. Here, the “category” seems to be defined by the original class label in the image database. One problem is that many natural images contain more than one type of object, but only one was labeled in each image. While the “apple” image in Figure 2o is used to argue that the neuron doesn’t encode the category “apple”, it is clear that the leftmost example image contains not only apples but also furniture, which also appears in other preferred images, weakening the claim that the neuron doesn’t encode object category.

We thank the reviewer for the expert and constructive comments.

The reviewer is correct that one of our main conclusions is that visual features explain the responses of MTL neurons better than human-labeled object categories. In this specific case mentioned by the reviewer, the category was defined by consensus judgment in the original image database, with one label per image, even when multiple objects were present. A key novelty of our findings is that, regardless of the dominant and consensus human labels, the neurons’ responses were better explained by the dominant visual features. Images with multiple objects indeed support our conclusion that the responses of MTL neurons align more closely with the dominant visual features (as clustered in visual feature space) than with the dominant human-defined object category labels.

Furthermore, we would like to take this opportunity to reiterate the evidence supporting that feature coding can serve as a more comprehensive mechanism to explain neural responses to visual objects.

First, feature coding does not presuppose any categorical membership of individual images as long as they share similar visual features (e.g., **Fig. 1d**; **Fig. 2l, m, o**; **Fig. 4d**). Second, when images are “misclassified” in terms of category membership (e.g., in **Fig. 2o**, where a more appropriate dominant object category might be “furniture” rather than “fruit” or “apple”), the neural responses still followed the dominant visual features. Third, when not all objects within a category are encoded, feature coding is favored over category coding (**Fig. 2n**; **Fig. 3a**). Therefore, feature selectivity can be dissociated from category selectivity in certain cases, and feature encoding does not depend on categories but solely on features. Importantly, feature coding explained the data better than category coding, and this was consistent across experiments.

2) Definition of memorability. Figure 5 shows a complex relationship between the memorability of an image, the relationship between the image and the neuron’s tuning region, and task performance. The authors defined memorability using a pre-trained DNN. Both DNN-estimated memorability and region-based neural preference were positively correlated with performance, but the two indices were negatively correlated (Figure 5L). This suggests a dissociation between memorability and performance, as otherwise they should have a similar relationship with the region-based neural preference. Since the task used here is basically a recognition memory task, and what it measures is the memorability of the image, its dissociation with the DNN estimate simply indicates that the estimated memorability is inaccurate. I think the DNN estimated scores here are quite confusing, and the authors could simply remove them. Otherwise, please clarify why the estimated scores differ from the subjects’ performance.

We thank the reviewer for pointing this out and apologize for any confusion caused by the presentation of our results. We would like to note that the results regarding memorability were not in conflict with those related to memory task performance. In the revised manuscript, we have clarified the methods, included a dedicated paragraph for motivation, and rewritten the results as follows.

First, we have clarified the memorability score in **Methods**:

“We used a pre-trained DNN model, ResMem [1], designed to predict the memorability of images, to calculate the memorability score for each image. The memorability score from ResMem represents how likely an image is to be remembered by people after a single view. The score ranges from 0 to 1, with

scores closer to 1 indicating that the image is highly memorable, meaning most people who see it are likely to remember it, and scores closer to 0 suggesting that the image is less memorable, meaning people are less likely to recall it after seeing it. This score is derived from patterns learned by the ResMem model, which was trained on large datasets of images with known memorability ratings. The model analyzes the visual features of an image and predicts how effectively it will be stored in and retrieved from human memory.”

Second, while memorability can predict memory performance (**Fig. 5c-f**), it is not the same as memory performance and may even be dissociated from it in certain cases, as the reviewer correctly pointed out (please see [2] for a comprehensive review of the cognitive processes involved in memorability). For this reason, and given that the human MTL is involved in encoding memorability [3], we examined how MTL neurons encoded memorability in addition to memory performance in this study. We have provided a better rationale for this analysis in the revised manuscript.

“Intrinsic image memorability—the likelihood that an image will be remembered by individuals after viewing it—consistently influences memory behavior across observers [1, 2]. Research has shown that certain visual properties make some images more memorable than others, regardless of personal significance or familiarity [1]. Factors such as distinctiveness, emotional impact, and meaningful content contribute to this memorability. Since memorability may not represent the same neurocognitive processes as explicitly reported memory performance [2], and given that encoding memorability actively engages the human MTL [3], we next investigated whether feature neurons were involved in encoding general image memorability, in addition to predicting memory performance. Advanced computational models, such as ResMem [1], have been developed to predict how memorable an image is based on its visual features. We employed this well-established pre-trained DNN implementation of memorability to calculate the memorability score for each image (**Methods**; see **Fig. 5a** for examples and **Fig. 5b** for the distribution of memorability scores for all stimuli).”

Third, we apologize for the way we presented our results, which caused some confusion. We observed a mix of two populations of neurons that were positively and negatively correlated with memorability scores (similar across all four experiments). Notably, both directions of correlation carry information, as typically seen in other studies (e.g., human MTL neurons can increase or decrease their firing rate to encode memory [4], fearful faces [5, 6], attended stimuli [7], familiarity [8], and social traits [9]).

Therefore, we expected to observe both neurons that exhibit a higher firing rate for more memorable stimuli and neurons that exhibit a higher firing rate for less memorable stimuli. Importantly, the key finding is that feature neurons had a greater percentage of neurons encoding memorability, suggesting that feature coding is related to memorability encoding. We have rewritten the results and revised **Fig. 5** accordingly:

“We next investigated whether feature neurons were more likely to encode image memorability during the learning phase (see **Supplementary Information** for analysis of the recognition phase; **Supplementary Fig. 5**). First, we observed a significant population of MTL neurons that correlated with levels of image memorability (Pearson correlation between firing rate and memorability score, $P < 0.05$; 104/1162, 8.95%, binomial $P = 7.44 \times 10^{-9}$; 47 neurons increased firing rate [**Fig. 5g, i**] and 57 neurons decreased firing rate [**Fig. 5h, j**] for higher memorability). Importantly, feature neurons had a significantly higher proportion encoding image memorability (31/116, 26.72%, binomial $P = 1.44 \times 10^{-15}$) compared to all recorded neurons (χ^2 -test between feature [$n_{\text{Feature\&Memorability}} / n_{\text{Feature}}$] and all [$n_{\text{Memorability}} / n_{\text{All}}$]; $P < 10^{-20}$; **Fig. 5k**), suggesting that feature neurons were more involved in encoding image memorability compared to non-feature neurons. Furthermore, 56 out of 116 feature neurons significantly differentiated memorability scores between in-region and out-region stimuli, suggesting that feature neurons’ tuning regions differentiated image memorability. It is worth noting that, based on previous memory studies [4], we expected to observe a mix of two populations of MTL neurons: one exhibiting a higher firing rate for more memorable stimuli and the other exhibiting a higher firing rate for less memorable stimuli. Both populations were considered to carry information and encode image memorability.

While, at the population level, in-region stimuli had a significantly lower memorability score compared to out-region stimuli (two-tailed paired t -test: $t(115) = 3.72$, $P = 0.0003$, $d = 0.51$, 95% CI = [0.014, 0.046]; note that 50/116 neurons had a higher memorability score for in-region stimuli), stimuli with lower memorability scores could still be better remembered (**Fig. 4e-g**) as long as they fell within the tuning regions of feature neurons. This result also suggests that feature neurons may play a role in enhancing the memory of stimuli within their regions. Indeed, we found that in-region stimuli had a significantly higher hit rate (**Fig. 5l**; two-tailed paired t -test: $t(64) = 2.60$, $P = 0.012$, $d = 0.39$, 95% CI = [0.01, 0.08]) and memory strength (**Fig. 5m**; $t(64) = 2.01$, $P = 0.049$, $d = 0.27$, 95% CI = [0.0007, 0.34]) compared to out-region stimuli that had a comparable memorability score (within $\pm 2\text{SD}$ of the mean of

the in-region stimuli). Together, we have shown that feature neurons are associated with encoding image memorability (see **Supplementary Information** for further comparison between feature neurons and memory-encoding neurons).

We next sought to address whether the above results generalized to tasks without an explicit memory component. To answer this question, we calculated the memorability scores for the ImageNet (**Fig. 1**) and COCO (**Fig. 3**) stimuli. For the ImageNet stimuli, we found that feature neurons had a significantly higher proportion of neurons that discriminated levels of image memorability (25/89, 28.09%, binomial $P = 1.38 \times 10^{-13}$), compared to all recorded neurons (77/874, 8.81%, binomial $P = 9.00 \times 10^{-7}$; χ^2 -test between feature and all: $P = 1.79 \times 10^{-8}$). We also found that 43 out of 89 feature neurons had a significantly different memorability score between in-region and out-region stimuli. For the COCO stimuli, again, we found that 43 out of 61 feature neurons had a significantly different memorability score between in-region and out-region stimuli. Therefore, feature neurons still encoded image memorability even when the tasks did not involve explicit memory decisions.

Finally, we replicated our findings with fixation-based analysis in the continuous recognition memory task (**Supplementary Information**).

Together, our results suggest that, in addition to memory performance, feature neurons with region-based coding are also more likely to encode memorability, both in tasks with and without explicitly reported memory performance.”

Fig. 5. Association between feature neurons and image memorability. **(a)** Example images with memorability scores shown on top. **(b)** Distribution of memorability scores across all stimuli used in the phased recognition memory task. **(c-f)** Image memorability predicted memory performance. **(c)** Accuracy (i.e., the proportion of correctly judged trials, including both hits and correct rejections). **(d)** Response level. **(e)** Reaction time (RT), relative to stimulus onset. **(f)** Hit rate (i.e., the proportion of trials correctly recognizing “old” stimuli). Each square represents a session, and error bars denote \pm SEM across sessions. Asterisks indicate a significant difference between high-memorability (top 30%) vs.

low-memorability (bottom 30%) stimuli using a two-tailed paired t -test. **: $P < 0.01$, ***: $P < 0.001$, and ****: $P < 0.0001$. **(g, h)** Example neurons that differentiated levels of image memorability. Trials are aligned to the stimulus onset. Shaded area denotes \pm SEM across trials. **(i, j)** The firing rate of these example neurons was correlated with image memorability scores. Each dot represents an image, and the gray line shows the linear fit. **(g, i)** Cell 1151. **(h, j)** Cell 1239. **(k)** The number of memorability-encoding neurons among feature neurons (left) and in the whole population (right). Purple: the number of memorability-encoding neurons ($n = 104$ for whole population and $n = 31$ for feature neurons). Green: the number of non-memorability-encoding feature neurons ($n = 85$). Gray: the number of non-memorability-encoding neurons ($n = 1058$). **(l, m)** Comparison of recognition performance for stimuli that fell within vs. outside feature neurons' tuning regions identified during the learning phase. Only out-region stimuli with a comparable memorability score as in-region stimuli (i.e., within ± 2 SD of the mean of the in-region stimuli) were analyzed. Legend conventions as in **Fig. 4**.

3) Memory modulation needs to be tested statistically. The title of Figure 6 is “Memory modulation of region-based feature coding”, but it seems that only the consistency of the preferred region between two phases has been statistically tested, not its modulation by task. The main evidence may be the 14 neurons with flexible coding of regions, but it's unclear whether the difference in estimated regions is simply due to noise in the data. This could be addressed by splitting the data into an earlier and a later half, and comparing the two estimated tuning regions.

We thank the reviewer for pointing this out. We would like to clarify that consistency and modulation are two sides of the same coin, both indexed by the percentage of overlap. To improve clarity, we have changed the title to: “Context dependency of region-based feature coding on memory”.

It is worth noting that we observed both highly consistent and highly flexible overlap in coding regions, suggesting that the flexibility in coding could not simply be attributed to noise in the data or temporal differences across sessions. To provide a positive control, we performed two analyses as suggested by the reviewer. First, we randomly subsampled 80% of the data from the learning phase twice (i.e., within session) and compared the distribution of the identified tuning regions. It is worth noting that splitting the data into earlier and later halves resulted in unstable coding regions because we needed more trials to reliably identify these regions (especially in this dataset where there were not many stimuli in the first

place). As expected, we observed high overlap of the tuning regions (the mean was $77.78\% \pm 22.29\%$ [mean \pm SD]; see **Supplementary Fig. 3j** for an example and **Supplementary Fig. 3k** for the group summary; cf. **Fig. 6e**). Second, we followed the procedure suggested by the reviewer (please refer to our response to Minor Question 6) to compare In vs. Out responses using training and testing stimuli from the same dataset. Again, we found that in-region stimuli elicited a significantly higher response than out-region stimuli (right-tailed paired t -test: $t(101) = 3.59$, $P = 2.56 \times 10^{-4}$). Together, these results have provided a positive control for cross-phase consistency.

Supplementary Fig. 3. Control analyses for region-based feature coding. **(d-i)** Control analyses with training and testing stimuli from the same dataset. Here, we used an 8-fold cross-validation procedure within the ImageNet and COCO datasets. Specifically, we split the data into 8 folds, using 7 of them to select the tuning region, and the remaining fold to test the response within vs. outside the region. This procedure was repeated 8 times, and all the feature neurons selected from the entire dataset were identified. We then performed the In vs. Out comparison on these feature neurons. We observed a significantly higher response to in-region stimuli compared to out-region stimuli in the ImageNet dataset (right-tailed paired t -test: $t(88) = 6.81$, $P = 5.76 \times 10^{-10}$), the COCO dataset ($t(60) = 5.34$, $P = 7.48 \times 10^{-7}$), and during the learning phase of the phased recognition memory task ($t(101) = 3.59$, $P = 2.56 \times 10^{-4}$). These results served as a positive control for cross-dataset generalization and cross-phase consistency. **(d, e)** ImageNet dataset. **(f, g)** COCO dataset. **(h, i)** Learning phase of the phased recognition memory task. **(e, g, i)** Population results comparing neuronal response to in-region testing stimuli vs. out-region testing stimuli. Legend conventions as in **Fig. 3d, e**. ***: $P < 0.001$, and ****: $P < 0.0001$. **(j, k)** Consistency of the coding regions during the learning phase of the phased recognition memory task. We randomly subsampled 80% of the data from the learning phase twice and compared the distribution of

the identified tuning regions. **(j)** An example neuron. **(k)** Group summary. Legend conventions as in **Fig. 6e**.

And in the main text:

“Among these feature neurons, 16 exhibited invariant coding across task phases (i.e., quantified as having more than 50% region overlap; see **Fig. 6c, d** for examples; see **Fig. 6e** for the distribution of the percentage of region overlap; see also **Supplementary Fig. 3h-k** for positive control analyses demonstrating consistency within tasks), while the remaining 14 exhibited more flexible coding of regions (**Fig. 6e**; see **Supplementary Fig. 2e, j** for a breakdown of amygdala and hippocampal neurons).”

Minor:

1)Page 5: “Feature Dimension 1 represented changes in object animacy”. The description seems inaccurate, as flowers appear on the positive side of dimension 1, along with animal images.

We agree with the reviewer. The first dimension of the feature space should be interpreted as artificiality. We have corrected this in the revised manuscript:

“Feature Dimension 1 primarily represented the transition from artificial to natural objects (consistent with [10]), while Feature Dimension 2 mostly captured variations in object size and the shift from indoor to outdoor objects.”

2)Page 5: “We then projected the response of a given neuron to each object onto the feature space by multiplying the response magnitude to each object to its corresponding location in the feature space, resulting in a response-weighted 2D feature map”. Since the objects occupy discrete locations in the feature space, how is a continuous 2D feature map obtained?

We thank the reviewer for asking this question. We estimated the continuous density map by smoothing the discrete firing rate map using a 2D Gaussian kernel (please see the section “*Selection of region-coding feature neurons*” in **Methods**: “To select feature neurons, we first estimated a continuous spike

density map in the feature space by smoothing the discrete firing rate map using a 2D Gaussian kernel.”). We have clarified this in the main text in the revised manuscript:

“To formally quantify the tuning of feature neurons (see **Methods; Supplementary Table 1**), we estimated a continuous spike density map in the 2D feature space (**Fig. 1c, d** upper right) by smoothing the discrete firing rate map (**Fig. 1c, d** middle) using a 2D Gaussian kernel and used a permutation test (1000 runs; **Fig. 1c, d** lower right) to identify the region(s) that had a significantly higher spike density above chance (red/cyan outlines in **Fig. 1c-e**; significant pixels were selected with permutation $P < 0.01$ and cluster size thresholds; similar results were derived with further cluster correction across adjacent significant pixels [11]).”

3)Page 5: *“Moreover, we found that only 3.20% (below-chance) of the amygdala and hippocampal neurons exhibited axis-based feature coding, consistent with our prior report examining neuronal responses to faces”. More details are needed. How are axis-based coding neurons defined?*

We thank the reviewer for pointing this out. We selected axis-coding neurons (i.e., neurons that encoded a linear combination of visual features) using partial least squares (PLS) regression with DNN feature maps [12, 13]. The detailed methods were described in the “*Regression analysis*” section of **Methods**, but we have now revised the title of this section to “*Selection of axis-coding neurons*” for greater clarity. We apologize for the confusion. We have also revised this section of methods as follows:

“*Selection of axis-coding neurons*”

To identify axis-coding neurons, i.e., neurons that encoded a linear combination of visual features, we employed a partial least squares (PLS) regression with DNN feature maps. This PLS method has been shown to be effective to study the neural response to DNN features [12, 13]. We used 10 components for each layer (explaining at least 80% of variance; we selected the number of components with a 10-fold cross validation to minimize the prediction error) and a permutation test with 1000 runs to determine whether a neuron encoded a significant axis-coding model. In each run, we randomly shuffled the object labels and used 50% of the objects as the training dataset. We used the training dataset to construct a model (i.e., deriving regression coefficients), predicted responses using this model for each object in the remaining 50% of objects (i.e., test dataset), and computed the Pearson correlation between the predicted

and actual response in the test dataset. The distribution of correlation coefficients computed *with shuffling* (i.e., null distribution) was eventually compared to the one *without shuffling* (i.e., observed response). If the correlation coefficient of the observed response was greater than 95% of the correlation coefficients from the null distribution, this axis-coding model was considered *significant*. This procedure has been shown to be very effective to select units with significant face models [14]. The correlation coefficient could also indicate the model's predictability and thus be compared between different neurons."

And in the main text:

"Moreover, we found that only 3.20% (below the 5% chance level) of the amygdala and hippocampal neurons exhibited axis-based feature coding (i.e., neurons encoding a linear combination of DNN visual features; see **Methods** for details), consistent with our prior report examining neuronal responses to faces [15]."

4)Page 7: "*whereas SC neurons showed steeper changes from the most- to the least preferred categories*". It's not so obvious from the figure alone. I understand that the difference between the top two images was quantified in Figure 2E, but not for other images.

We thank the reviewer for pointing this out. The reviewer is correct that **Fig. 2e** further illustrates the difference between the top two images in **Fig. 2d**. We highlighted this difference because it is particularly important for SC neurons, which have a single most-preferred object category, but it was difficult to clearly observe in **Fig. 2d**. The other images *collectively* demonstrate that both MC and SC neurons show steeper changes from their most- to least-preferred categories compared to non-category-selective neurons, with MC neurons displaying even steeper changes in the less-preferred categories compared to SC neurons (right part of the curve). These results are clearly visible in **Fig. 2d**, and it is not necessary to show the comparison for each *individual* category in the ranking.

5)Page 7: “In addition, single-trial population decoding revealed that MC neurons differentiated between more stimuli than SC neurons”. It’s unclear whether the number of neurons was matched between two groups of neurons.

We thank the reviewer for pointing this out. We have conducted an additional decoding analysis where we subsampled MC neurons with the same number of SC neurons for 1000 times, in order to match the number of neurons between groups. The result consistently showed that MC neurons achieved better performance on decoding than SC neurons (**Revision Fig. 1**).

Revision Fig. 1. Decoding analysis with an equal number of SC and MC neurons. Legend conventions as in **Fig. 2h**.

We have included the following results in the revised manuscript:

“In addition, single-trial population decoding revealed that MC neurons differentiated between more stimuli than SC neurons (Fig. 2g, h; note that similar results were derived when we matched the number of neurons between groups).”

6)The example cell in Figure 3D is not very compelling. Where is this cell in Figure 3E? It would be nice to see the plot as in Figure 3E, but comparing COCO stimuli within and outside the region, as a positive control. For example, one could use half of the COCO data to estimate the tuning region, and the other half to compute the mean response within and outside the region.

We thank the reviewer for this great suggestion. In the revised manuscript, we validated our results using an 8-fold cross-validation procedure *within* the ImageNet and COCO datasets, as suggested by the reviewer. Specifically, we split the data into 8 folds, using 7 of them to select the tuning region, and the remaining fold to test the response within vs. outside the region (it is worth noting that more trials are needed to reliably identify coding regions than to conduct testing). This procedure was repeated 8 times, and all the feature neurons selected from the entire dataset were identified. We then performed the In vs. Out comparison on these feature neurons. As expected, our results revealed a robust difference in response between the In vs. Out stimuli for both the ImageNet dataset (**Supplementary Fig. 3d, e**; right-tailed paired t -test: $t(88) = 6.81$, $P = 5.76 \times 10^{-10}$) and the COCO dataset (**Supplementary Fig. 3f, g**; $t(60) = 5.34$, $P = 7.48 \times 10^{-7}$). These results served as a positive control for the cross-dataset generalization.

Supplementary Fig. 3. Control analyses for region-based feature coding. **(d-i)** Control analyses with training and testing stimuli from the same dataset. Here, we used an 8-fold cross-validation procedure within the ImageNet and COCO datasets. Specifically, we split the data into 8 folds, using 7 of them to select the tuning region, and the remaining fold to test the response within vs. outside the region. This procedure was repeated 8 times, and all the feature neurons selected from the entire dataset were identified. We then performed the In vs. Out comparison on these feature neurons. We observed a significantly higher response to in-region stimuli compared to out-region stimuli in the ImageNet dataset (right-tailed paired t -test: $t(88) = 6.81$, $P = 5.76 \times 10^{-10}$), the COCO dataset ($t(60) = 5.34$, $P = 7.48 \times 10^{-7}$), and during the learning phase of the phased recognition memory task ($t(101) = 3.59$, $P = 2.56 \times 10^{-4}$). These results served as a positive control for cross-dataset generalization and cross-phase consistency. **(d, e)** ImageNet dataset. **(f, g)** COCO dataset. **(h, i)** Learning phase of the phased recognition memory

task. **(e, g, i)** Population results comparing neuronal response to in-region testing stimuli vs. out-region testing stimuli. Legend conventions as in **Fig. 3d, e**. ***: $P < 0.001$, and ****: $P < 0.0001$.

And:

“Our results confirmed this hypothesis (right-tailed paired t -test: $t(21) = 1.81$, $P = 0.04$, $d = 0.16$, 95% CI = $[0.02, \infty]$; see **Fig. 3d** for an example and **Fig. 3e** for group results; see **Supplementary Fig. 3c-f** for control analyses with training and testing stimuli from the same dataset [**Supplementary Fig. 3c, d** for the ImageNet stimuli and **Supplementary Fig. 3e, f** for the COCO stimuli]), suggesting that region-based feature tuning generalized between different image sets.”

We have also labeled the example cell in the revised **Fig. 3e** (see below).

Fig. 3. (e) Population results comparing neuronal response to ImageNet stimuli falling in vs. out of the tuning region ($n = 22$). Each dot represents a neuron. Error bars denote \pm SEM across neurons. Asterisks indicate a significant difference between the In vs. Out responses using a right-tailed paired t -test ($P < 0.05$). The example neuron shown in **(d)** is highlighted in a darker color.

7)Page 14: “we observed that feature neurons predicted memory performance, indicating their distinct involvement in memory processes compared to category-selective neurons”. Where are the results for category-selective neurons?

We thank the reviewer for pointing this out. The results for category-selective neurons were described in the previous study [16], which showed that category-selective neurons function orthogonally to memory neurons. In this study, we analyzed the same data and found that feature neurons could predict memory performance. We have clarified this in the revised manuscript:

“Furthermore, unlike previous findings that showed category-selective neurons to be orthogonal to memory neurons (i.e., they are insensitive to whether a stimulus is novel or familiar or whether a stimulus is retrieved with high or low confidence) [16], using the same data, we observed that feature neurons predicted memory performance (Fig. 4), indicating their distinct involvement in memory processes compared to category-selective neurons (details shown in [16]).”

Reply to comments from Reviewer 2

I reviewed this manuscript previously for another journal and my opinion remains largely the same. This manuscript describes results from experiments on visual perception and memory using single-unit recordings of the human hippocampus and amygdala in subjects with epilepsy undergoing invasive EEG. The main goals were to identify how hippocampal/amygdala neurons code for high-level stimulus features and how this is related to memory. A neural network model of visual perception was used to project a large set of visual stimuli into a 2-dimensional feature space, including dimensions for object animacy and object size. About 10% of identified hippocampal/amygdala units responded maximally to a particular range of this 2-dimensional feature space, which is described as a “region-based feature code.” A number of control analyses were performed to validate this coding across different stimulus sets and to test accuracy in predicting neural activity relative to category-selective coding. In a recognition memory experiment, stimuli that fell within the feature tuning region of neurons showing the region-based feature code were better recognized with higher confidence than stimuli that fell outside the tuning region of identified neurons. The authors also investigated “memorability”, which refers to the tendency for some stimuli to be recognized consistently across individuals in memory tests, although the findings were somewhat less clear (feature neurons discriminated stimulus memorability better than other neurons, but, counterintuitively, memorability was lower for stimuli that were in-region for feature-sensitive neurons despite better memory performance for these stimuli). Finally, by comparing feature-sensitivity across encoding versus retrieval phases of the task, the authors found that many neurons changed the region of the feature space to which they were maximally selective, whereas other neurons remained consistent, demonstrating memory modulation.

Although many technical aspects of this report are impressive, including a difficult-to-acquire dataset and a number of sophisticated analyses that blend neural network modeling and neural response modeling, I find it difficult to find any major significance in the results. Although the authors make claims about how the findings fundamentally inform understanding of the perceptual-memory transition in hippocampus/amygdala, I consider the impact for perception as subtle, and the link to memory weak and confusing. The main limitation of the report is that it is primarily descriptive and does not include experimental tests needed to support the major conclusions. That is, the authors look for general relationships between feature coding and memory performance but without including relevant manipulations within the task (or within the brain) to test hypotheses about the nature of these

relationships. Instead, the general patterns identified in the data motivate hypotheses about brain-perception-memory relationships, which are presented as conclusions.

We thank the reviewer for the expert and constructive comments. The reviewer had an accurate summary of our results.

First, it is important to highlight the novelty and significance of our study: we have introduced a region-based feature code that has not been previously documented in the literature. In fact, it challenges the prevailing notion of feature-invariant exemplar-based coding in the amygdala and hippocampus [17-19]. This region-based feature code explains the long-standing visual category selectivity observed in the human amygdala and hippocampus and constitutes a more general mechanism of visual tuning for neurons in these regions. Furthermore, this region-based feature code is not only related to memory performance and memorability but also proposes a clear mechanistic hypothesis for the perceptual-memory transition, which warrants further exploration in future studies (see below for more details). While our study is correlational (as the reviewer correctly pointed out), like the vast majority of human studies, the general patterns identified in our data motivate important hypotheses about brain-perception-memory relationships, and our data provide the first evidence to support these hypotheses.

Second, we employed a common and effective procedure to examine neural responses to perceptual features. Previous studies using this procedure have revealed visual category selectivity in the human MTL [20] and axis coding in the monkey IT cortex [10, 14]. In this study, we applied this established method and uncovered a novel perceptual code (e.g., region-based feature coding) in the human MTL.

Third, we employed a highly established procedure to study the neural correlates of memory by comparing remembered vs. forgotten stimuli [21, 22], which is the relevant manipulation within the task. This approach is widely used to investigate the relationship between neural activity and memory performance. Specifically, by examining the differential neuronal responses to remembered and forgotten stimuli, we directly tested how feature coding relates to memory processes. Therefore, while we did not introduce manipulations within the brain (e.g., neurostimulation), our experimental design follows well-validated methods to explore these relationships during recognition memory and includes the key experimental variable/manipulation to investigate memory processes.

Fourth, it is worth noting that the identical task (and a subset of the present data) has been successfully used to delineate the *correlational* relationships between visual category selectivity (not feature coding) and memory performance [16]. Therefore, our data is well-suited to address our hypothesis-driven questions about the relationship between perception and memory. However, to more clearly and accurately describe our results, we have removed all statements implying a causal relationship between perception and memory (please see below).

Fifth, we have provided more details about how perceptual representations of visual features are translated into a region code for object categories. We have also moderated our arguments regarding the transition mechanism. Please refer to our response below.

Lastly, we have substantially revised our description of the memorability results and better motivated this section of results. Please refer to our response to Reviewer 1's Question 2.

1. Region-based feature coding of visual objects and category selectivity: A major conclusion is that the identified region-based feature code can “explain”, or is more fundamental than, the category/exemplar-based coding that had previously been identified in these brain areas. However, this conclusion is not fully warranted, as this pattern was the case for neurons that were sensitive to multiple categories (MC), but not those that were sensitive to single categories (SC). A challenge for disentangling these two coding schemes is that feature space and categories are largely correlated, and without specific attempts to de-correlate them or experimental methods to systematically explore divergence, the findings add nuance to the question of how these neural populations code visual features, without being able to strongly support one model over another.

We thank the reviewer for raising these important questions. First, it is worth noting that single categories can also be explained within our region-coding framework. In comparison to multiple categories, single categories exhibit a narrower tuning region. Second, the reviewer is correct in noting the similarity between the organization of visual feature space and categories, which aligns with expectations based on brain representations. However, it is important to note that our objective was not to disassociate feature space from categories; rather, we aimed to introduce a novel approach for more comprehensively describing the longstanding category-selectivity phenomenon observed in the

amygdala and hippocampus. As outlined in the **Discussion** section, our region-based feature code offers a deeper understanding of the neural response to visual objects. This approach offers several clear advantages, including its ability to explain data not accounted for by traditional categorical models (such as the response to a subset of specific objects within a category or the misclassification of objects), without relying on predefined categorical membership. Importantly, this novel perspective has not been fully recognized in existing literature. On the other hand, we agree with the reviewer that it would be valuable to conduct further experiments in cases where features and categories can be more effectively disentangled.

More fundamentally, however, the claims that the findings identify “how the transformation from a perception-driven representation to the memory-based semantic representation in the MTL is achieved” are misplaced, as the study did not compare coding schemes along the visual hierarchy, and did not offer a definition of “semantic representations” with enough specificity to be tested. Qualitative comparisons of how small subsets of neurons respond to visual features among different experiments investigating different brain regions are insufficient for addressing these important questions.

We thank the reviewer for this important question. While the primary focus of our study is to characterize a novel region-based feature code and delineate its relationships with category selectivity and memory, this neural code can indeed link dense representations of visual features and sparse representations of categories in the MTL. Specifically, neurons in the primate IT cortex—referred to as the ventral temporal cortex (VTC) in humans—encode complex visual features in a parametric manner, exhibiting axis coding [10, 13, 14, 23-26]. These IT/VTC neurons can form a neural feature space, with the axes of this space encoded by axis-coding neurons (**Revision Fig. 2**). MTL neurons, which receive processed visual information from the IT/VTC, particularly the axes of the neural feature space, encode a receptive field (i.e., a coding region) within this space, becoming selective to identities within that receptive field (**Revision Fig. 2**). Thus, region-based coding provides a computational framework that bridges IT/VTC and MTL representations, enabling us to understand how perceptual representations in the IT/VTC are translated into memory-based semantic representations in the MTL (note that the category-selective neurons have been hypothesized to represent semantic memories [27], which in turn form the foundation of declarative memories [18]).

Revision Fig. 2. A neural computational framework explains the transition from dense feature-based coding in the ventral temporal cortex (VTC) to sparse semantic-based coding in the medial temporal lobe (MTL). Neurons/iEEG channels in the VTC represent visual features using an axis code, forming neural coding axes that collectively create a neural feature space. MTL neurons receive processed visual input from the VTC and encode a receptive field (i.e., coding region) within this neural feature space, making them selective to stimuli that fall within this receptive field. As a result, MTL neurons exhibit region coding.

To test this hypothesis, we examined whether the same region-coding neurons in this study could encode regions within the neural feature space. Using the same stimuli and simultaneously recorded intracranial EEG (iEEG), we first demonstrated that iEEG channels in the VTC (fusiform gyrus [FG] and inferotemporal cortex [IT]) exhibited axis coding (see **Revision Fig. 3a, b** for example channels and **Revision Fig. 3c-e** for group summary). With these neural axes, we constructed a neural feature space

(**Revision Fig. 4a**). Notably, this neural feature space had a highly organized structure, suggesting that it contained sufficient visual information for object coding. Importantly, we observed that MTL neurons exhibited region coding within this neural feature space (see **Revision Fig. 4b, c** for example channels and **Revision Fig. 4d-i** for group summary; legend conventions as in **Fig. 2**), strongly supporting our hypothesis of the transition mechanism.

Revision Fig. 3. Axis-based feature coding in the ventral temporal cortex (VTC). (**a-b**) Example channels exhibiting axis coding. (**a**) Fusiform gyrus (FG). (**b**) Inferior temporal gyrus (IT). (left) The z-scored high-gamma power (HGP) changed as a linear function of the first partial least squares (PLS) component of the feature map. Each dot represents an object image and color coding denotes the object category. The gray line represents the linear fit. (right) Visualization of the encoded feature axis. Note that both channels showed a significant relationship with the feature map (PLS regression, permutation $P < 0.001$). (**c**) Percentage of channels exhibiting axis coding. Dark colors represent an above-chance number of selected channels in the corresponding ROIs (binomial test: $P < 0.05$; Bonferroni correction for multiple comparisons), while light colors indicate chance-level selection. χ^2 -test was performed to

compare the proportion of axis-coding channels between the left and right hemispheres. **: $P < 0.01$. PH: posterior hippocampus. AH: anterior hippocampus. **(d)** Distribution of axis-coding channels in the VTC and MTL. Color coding shows the strength of axis coding (Pearson's r between observed and predicted response). **(e)** Strength of axis coding (Pearson's r) averaged across all visually responsive channels in each ROI. Dark colors represent significant ROIs (right-tailed one-sample t -test against 0: $P < 0.05$). Two-tailed two-sample t -test was performed to compare the strength of axis coding between the left and right hemispheres. **: $P < 0.01$.

Revision Fig. 4. Region-based feature coding of MTL neurons in the VTC neural feature space. **(a)** VTC neural feature space constructed using all axis-coding channels in the FG. All stimuli are shown in this space. **(b, c)** Two example MTL neurons that encoded a region containing visually similar object images in the VTC neural feature space. The left panel shows the neuronal responses to 500 objects (50 object categories). Trials are aligned to stimulus onset (gray line) and are grouped by individual object category. On each box, the central mark is the median, the edges of the box are the 25th and 75th percentiles, the whiskers extend to the most extreme data points the algorithm considers to be not outliers, and the circles denote the outliers. The middle panel shows the projection of the firing rate onto

the feature space. Each color represents a different object category. The size of the dot indicates the firing rate. The right panel shows the estimate of the spike density in the feature space. By comparing observed (upper) vs. permuted (lower) responses, we could identify a region where the observed neuronal response was significantly higher in the feature space. This region was defined as the tuning region of a neuron (delineated by the red/cyan outlines; also shown in **(a)**). **(d-i)** Population summary of MTL region-coding neurons. **(d)** Percentage of region-coding neurons in each MTL ROI. **(e)** The number of categories encoded by region-coding neurons. **(f)** The number of objects encoded by region-coding neurons (i.e., the number of object images that fell within the tuning region of a region-coding neuron). **(g-i)** The population aggregated tuning region in each MTL ROI. Color bars show the counts of overlap between individual tuning regions. Numbers in the density maps show the percentage of the VTC neural feature space covered by the tuning regions of the total observed region-coding neurons.

Given the focus of the present study, we have decided to present these results in a future study. It is worth noting that we have already downplayed the transition mechanisms in the current manuscript. Additionally, we have clarified the representation of semantics (i.e., sparse coding of object categories) in the revised manuscript.

2. Region-based feature coding and memory: A major conclusion is that “feature neurons” (neurons that show selectivity to a region of feature space) “play a crucial role in memory processing.” The evidence for this is that stimuli falling into the feature space that was preferred by the recorded neurons were better recognized than those that were not. I am honestly at a loss to understand what this means about how feature neurons contribute to memory, and I do not find that any clear mechanistic explanation was provided.

One point that is very confusing is that only a small subset of the entire population of neurons were recorded. What is the rationale for there being a relationship between stimuli that happen to fall into the feature space of the small subset of neurons that happened to be recorded and memory performance? One possibility is that the portions of feature space coded by feature neurons are particularly salient or have some other special perceptual or semantic property. This could impact perceptual coding and memory, not necessarily through the same mechanism. The findings regarding memorability are equally puzzling, as the relationship is not what would have been expected based on the relationship between

feature coding and memory, and the interpretation is very post hoc. The authors should provide a mechanistic test of the relationship between feature neuron coding and memory as the correlation reported is hard to interpret.

We thank the reviewer for the opportunity to clarify the rationale behind this analysis, which was based on two prominent features of MTL neurons and strongly grounded in theory. First, category-selective sparse-coding neurons are considered fundamental to declarative memory [17, 18, 28, 29]. Region-coding neurons, which can explain category selectivity, also exhibit sparse coding properties. Therefore, it is important to determine whether region-coding neurons are associated with aspects of declarative memory. The most natural comparison is between remembered and forgotten stimuli, as shown in previous literature [30]. Second, some stimuli are remembered better than others, likely due to their visual features or saliency (captured by memorability; see our response to Reviewer 1's Question 2). Since region-based feature coding is a perceptual code, it is logical to explore whether perceptually encoded stimuli (in-region stimuli) are preferentially encoded into memory. While we only sampled a small subset of neurons, we could still estimate which stimuli were preferentially encoded, as done in established analyses of category selectivity [16, 20, 31]. It is important to note that prior studies have used the same correlational analysis to investigate the relationship between category selectivity and memory encoding [16].

We have better motivated this analysis in the revised manuscript:

“Above, we demonstrated that region-based feature coding can serve as a more comprehensive framework to explain the classical category-specific neural response to objects in the human MTL. Category-selective sparse-coding neurons are considered the building blocks of declarative memory [17, 18, 28, 29]. However, it remains unclear whether region-coding feature neurons, which not only explain category selectivity but also exhibit sparse coding properties, are linked to aspects of declarative memory, particularly when comparing remembered vs. forgotten stimuli, as shown in prior research [30]. Moreover, some stimuli are more easily remembered due to their inherent perceptual features or saliency. Given that region-based feature coding is fundamentally a perceptual process, we hypothesize that stimuli encoded through this mechanism (in-region stimuli) may be preferentially stored in memory. To address these questions, we next investigated the contribution of region-based coding in feature neurons to memory performance using a phased recognition memory task [32] (Fig. 4a; see **Methods**).”

We apologize for our previous statement, “*Therefore, feature neurons play a crucial role in memory processing,*” which was inaccurate. We agree with the reviewer that this statement implies a mechanistic/causal relationship between feature coding and memory, which is not supported by the present study. We have now revised the discussion accordingly and ensured that statements about these results throughout the manuscript are accurate.

“In this study, we demonstrated that region-coding feature neurons were linked to aspects of declarative memory. Specifically, we found that stimuli within these neurons’ tuning regions were not only better remembered but also associated with greater memory strength (Fig. 4), suggesting that stimuli encoded through region-based feature coding may be preferentially stored in memory. Furthermore, region-coding feature neurons were more likely to encode memorability, both in tasks with and without explicitly reported memory performance (Fig. 5), indicating that region coding may effectively connect perceptual features to memory.”

We have also revised the Abstract and removed “*critical link between visual feature processing and semantic representations in memory*”.

Lastly, we agree with the reviewer that stimulus features (and thus the proportions of the feature space coded by feature neurons) may impact both perceptual coding and memory. For this reason, we investigated memorability (see our response to Reviewer 1’s Question 2). Although our results are correlational, they provide important initial evidence for future mechanistic testing of brain-perception-memory relationships.

3. For the main analysis showing that amygdala and hippocampal neurons exhibit region-based feature coding, only 10% of neurons out of the total of 1204 recorded demonstrated region-based feature coding, with a similar value in the replication experiment. Is this relatively low value predicted by any formal model or a priori considerations? Is there any alternative hypothesis? If not, how is this to be interpreted, in the context of ~90% of neurons not contributing to this function?

We thank the reviewer for pointing this out. We would like to clarify that, consistent with all previous studies [30], it is typical for only a small fraction of MTL neurons to encode a specific task aspect, such as region-based feature coding. This is expected given the wide range of cognitive functions the human

MTL supports, including memory [4], emotion [5, 6], attention [7], familiarity [8], spatial navigation [33], and social traits [9]. Therefore, the finding that ~10% of neurons demonstrated region-based feature coding aligns with this expectation and the results of similar studies, which report only a fraction of neurons involved in a particular task.

While region-based feature coding is a novel model, its proportion can be predicted by and is consistent with the category selectivity model (approximately 10% to 15%) [7, 16, 20] and region-based feature coding of faces (approximately 10%) [15]. Here, we tested the null hypothesis that this proportion could arise by chance using a binomial test (please see our response to Reviewer 3's Question 1). The observed proportion was significantly above chance (leading to the rejection of the null hypothesis), suggesting that a meaningful population of neurons is involved in region-based feature coding. It is also worth noting that in this study, neurons were not prescreened for region-based coding, meaning we sampled all neurons without bias. As a result, we anticipated that only a subset of neurons would be involved in this specific function, while others may contribute to different aspects, such as memory or stimulus novelty (**Fig. 4**) and image memorability (**Fig. 5**).

In summary, the proportion (~10%) of neurons contributing to region-based coding is expected and should be interpreted in the context of the MTL's broader functional diversity.

4. The authors make conclusions about region-based feature coding reflecting “semantic representations”, but there is no operational definition of this term given with sufficient clarity to evaluate this conclusion versus alternatives. There was no experimental manipulation to test whether the encoding processing was “semantic” versus of some other quality. Receptivity to a specific region of a 2D model of image features does not necessarily imply semantic representations. Moreover, some region-based feature coding neurons were specific to different features in different categories, which is opposite to what one would expect if they reflect semantic information processing, which should tend to be organized by category. In general, there are many alternatives, including that neurons may simply respond to stimuli with a certain range of visual complexity. Firm conclusions about function require specification of alternatives, and some modeling or tests sufficient to compare the preferred interpretation against these alternatives.

We thank the reviewer for pointing this out and apologize for the confusion. In our context, *semantic representation* and *semantic information processing* refer to object category coding (the reviewer is correct that semantic representation is typically organized by category). In contrast, region-based feature coding is based on visual features, which is broader than category-selective coding. This is because, in addition to encoding objects from the same category that share visual features, it can also encode different features across categories, as the reviewer correctly pointed out. We have clarified this throughout the manuscript.

Furthermore, unlike the representations in IT/VTC, where neurons encode complex visual features in a parametric manner and exhibit axis coding [10, 13, 14, 23-26], region-coding neurons encode a region within the high-level feature space, making them responsive to stimuli that fall within this region (please see above for our description of the transition mechanism). Therefore, region-based feature coding may serve as the basis for object representations (i.e., semantic representations) in the MTL.

Reply to comments from Reviewer 3

Sure, here's the revised version without subtitles:

The paper presents an intriguing study on electrophysiological responses from the amygdala and hippocampus to thousands of natural images under different tasks. The authors demonstrate that neurons' responses can be described as region-based feature codes, akin to receptive fields in a feature space. Additionally, they found that region-based feature-coded neuron responses better predict image memorability compared to non-region-based feature neurons. While the paper offers a novel perspective on the coding scheme, several details regarding the analysis require clarification. Here are my comments:

The paper states that 10% of neurons exhibit a regional coding scheme. However, it is unclear what coding scheme the remaining 90% of neurons follow. If the majority of neurons (90%) are not region-based coding, what alternative coding schemes do they utilize? Additionally, the binomial test referenced in Figure 1 lacks clarity. What is the null hypothesis of this test? It would be beneficial to include this information in the methods section.

We thank the reviewer for the expert and constructive comments.

We would like to clarify that, consistent with all previous studies [30], only a small fraction of MTL neurons encode a specific task aspect (e.g., region-based coding). Given the broad range of functions the human MTL is involved in, the remaining MTL neurons may encode other task-related aspects, such as memory [4], emotion [5, 6], attention [7], familiarity [8], spatial navigation [33], and social traits [9]. In this study, in addition to region-based coding, other neurons may encode memory or stimulus novelty (**Fig. 4**) and image memorability (**Fig. 5**). It is important to note that we did not prescreen neurons for region-based coding; thus, it is expected that only a small fraction of all recorded neurons would be involved in region-based coding. However, this approach allows for unbiased sampling of the proportion of neurons involved in a specific task aspect. We then used a binomial test (please see below) to determine whether the observed population is above chance. We have clarified the binomial test and the null hypothesis in the revised manuscript:

“Binomial test

We used the binomial test to determine whether the observed proportion of selected neurons differed significantly from a given chance level. Specifically, the null hypothesis assumed that the selection of neurons was due to chance (5%), while the alternative hypothesis suggested that the selection was greater than chance. The binomial test calculates the probability of observing at least k selected neurons out of a total of n neurons, given the probability p , using the cumulative distribution function (CDF) of the binomial distribution. The p-value represents the probability of obtaining the observed number of selected neurons (or more) under the null hypothesis:

$$\text{binomial test p-value} = P(X \geq k) = 1 - \sum_{i=0}^{k-1} \binom{n}{i} p^i (1-p)^{n-i}$$

This approach allows us to statistically assess whether the observed number of selected neurons is likely due to a real effect rather than random chance.”

The authors claim that only 3.87% of neurons exhibit the axis model. What criteria are used to accept or reject a neuron as following the axis model or the region-based model? The statement "below chance performance" is ambiguous. What is the chance performance of the axis model, and how is it determined? A detailed description of this process should be included in the methods section.

We thank the reviewer for pointing this out. The detailed methods, criteria, and parameters used to define axis-coding and region-coding neurons are described in the “*Selection of axis-coding neurons*” and “*Selection of region-coding feature neurons*” sections in **Methods**, respectively. We apologize that the previous section titles were not sufficiently informative for these methods.

Furthermore, the chance performance of the axis-coding model was 5%, as determined by the permutation test of the partial least squares (PLS) regression. A detailed description of the selection process has been included in the manuscript, and we have further clarified these methods in the revised version. Please refer to our response to Reviewer 1’s Minor Question 3 for more information.

Lastly, we have clarified this in the revised main text:

“Moreover, we found that only 3.20% (below the 5% chance level) of the amygdala and hippocampal neurons exhibited axis-based feature coding (i.e., neurons encoding a linear combination of DNN visual

features; see **Methods** for details), consistent with our prior report examining neuronal responses to faces [15].”

On page 11, in the first paragraph, it is mentioned that 31/116 (26.72%) neurons are feature neurons. However, the paper records 1162 neurons, not 116, and the proportion calculation appears incorrect. Please verify the numbers and correct the proportion accordingly. Additionally, the definition of “whole” in the context of “whole, 104/1162” requires clarification.

We thank the reviewer for pointing this out. We would like to clarify that in this analysis, we aimed to show that among feature neurons (116 neurons), a higher percentage encoded image memorability (31/116; 26.72%) compared to all recorded neurons (i.e., the “whole population” of 1162 neurons, of which only 104/1162 [8.95%] encoded image memorability). This observation was statistically confirmed using a χ^2 -test ($[n_{\text{Feature\&Memorability}} / n_{\text{Feature}}]$ vs. $[n_{\text{Memorability}} / n_{\text{All}}]$). We have substantially revised this section of the results in the revised manuscript (please also refer to our response to Reviewer 1’s Question 2). All the numbers and proportions have been verified and confirmed to be correct.

The feature space appears to be determined by the stimulus set used (PCA and t-SNE based on responses to the images). It would strengthen the paper to include analyses demonstrating that the feature space is stable across different image datasets, such as COCO images and ImageNet images.

We thank the reviewer for the suggestions. The reviewer is correct that the feature space is determined by the stimulus set used. Here, we demonstrated the stability of the feature spaces by comparing the individual ImageNet or COCO feature space to their combined feature space. First, qualitatively, both the ImageNet (**Fig. 1e**) and COCO (**Fig. 3b**) feature spaces, as well as their combined feature space (**Supplementary Fig. 3a**), captured changes in artificiality and animacy, suggesting that similar visual information was represented across different constructions. Second, quantitatively, we conducted a representational similarity analysis (RSA) between an individual feature space (i.e., ImageNet or COCO) and the combined feature space by calculating stimulus-by-stimulus correlations of feature space coordinates. Using a permutation test, we found that the stimulus representations (i.e., stimulus coordinates in the feature space) in both the ImageNet (**Supplementary Fig. 3b**; $P < 0.001$) and COCO

(**Supplementary Fig. 3c**; $P < 0.001$) feature spaces were significantly correlated with the stimulus representations in the combined feature space, suggesting that stimulus representations remained stable across different constructions of feature spaces and that the feature spaces were consistent across different image datasets. It is important to note that since the construction of a feature space depends on the specific stimuli used, we could not directly project the COCO stimuli onto the ImageNet feature space without using them in constructing that feature space, and vice versa for the ImageNet stimuli in the COCO feature space. Therefore, although the ImageNet and COCO feature spaces represent similar information, we could not directly compare them quantitatively. For this reason, we compared the individual feature spaces of ImageNet and COCO to their combined feature space, where different stimuli were used in constructing these spaces.

Supplementary Fig. 3. Control analyses for region-based feature coding. **(a-c)** Stability of the feature space across different constructions. **(a)** Feature space constructed using combined ImageNet and COCO stimuli. **(b, c)** Representational similarity analysis (RSA) between an individual feature space and the combined feature space. Stimulus-by-stimulus representational dissimilarity matrix (RDM) was calculated using stimulus coordinates from an individual feature space or the combined feature space. Using a permutation test, we found that the stimulus representations (i.e., stimulus coordinates in the feature space) in both the ImageNet ($P < 0.001$) and COCO ($P < 0.001$) feature spaces were significantly correlated with the stimulus representations in the combined feature space, suggesting that stimulus

representations remained stable across different constructions of feature spaces and that the feature spaces were consistent across different image datasets. **(b)** ImageNet. **(c)** COCO.

We have included the following results in the revised manuscript:

“In the common feature space for the ImageNet and COCO stimuli (see **Supplementary Fig. 3a-c** for stability of stimulus representations across different constructions of feature spaces), the tuning regions of 22 feature neurons selected using the COCO stimuli encompassed the ImageNet stimuli.”

The abstract and discussion emphasize recording over 3k neurons. However, it is noted that most neurons are not included in the analysis. To avoid potential misleading implications, it may be advisable to adjust the emphasis on the total number of recorded neurons.

We thank the reviewer for pointing this out. We would like to clarify that, as is common practice, the total number of neurons (3173 in this study) is typically reported (e.g., [5, 34-36]), as it provides a comparable illustration of sample size with other studies. Furthermore, as mentioned in response to the first question, all recorded neurons were included in the analysis, although only a small fraction of these neurons were selective to a specific task aspect (e.g., region-based coding). Notably, different task aspects (e.g., memory encoding and image memorability) were analyzed using the entire dataset. Therefore, reporting the total number of neurons is more informative.

References

1. Needell, C.D. and W.A. Bainbridge, *Embracing New Techniques in Deep Learning for Estimating Image Memorability*. Computational Brain & Behavior, 2022. **5**(2): p. 168-184.
2. Bainbridge, W.A., *The resiliency of image memorability: A predictor of memory separate from attention and priming*. Neuropsychologia, 2020. **141**: p. 107408.
3. Lahner, B., et al., *Visual perception of highly memorable images is mediated by a distributed network of ventral visual regions that enable a late memorability response*. PLOS Biology, 2024. **22**(4): p. e3002564.
4. Rutishauser, U., E.M. Schuman, and A.N. Mamelak, *Activity of human hippocampal and amygdala neurons during retrieval of declarative memories*. Proceedings of the National Academy of Sciences, 2008. **105**(1): p. 329-334.
5. Wang, S., et al., *Neurons in the human amygdala selective for perceived emotion*. Proceedings of the National Academy of Sciences, 2014. **111**(30): p. E3110-E3119.
6. Wang, S., et al., *The human amygdala parametrically encodes the intensity of specific facial emotions and their categorical ambiguity*. Nature Communications, 2017. **8**: p. 14821.
7. Wang, S., et al., *Encoding of Target Detection during Visual Search by Single Neurons in the Human Brain*. Current Biology, 2018. **28**(13): p. 2058-2069.e4.
8. Cao, R., et al., *Neural mechanisms of face familiarity and learning in the human amygdala and hippocampus*. Cell Reports, 2024. **43**(1): p. 113520.
9. Cao, R., et al., *A neuronal social trait space for first impressions in the human amygdala and hippocampus*. Molecular Psychiatry, 2022. **27**(8): p. 3501-3509.
10. Bao, P., et al., *A map of object space in primate inferotemporal cortex*. Nature, 2020. **583**(7814): p. 103-108.
11. Maris, E. and R. Oostenveld, *Nonparametric statistical testing of EEG- and MEG-data*. Journal of Neuroscience Methods, 2007. **164**(1): p. 177-190.
12. Yamins, D.L.K., et al., *Performance-optimized hierarchical models predict neural responses in higher visual cortex*. Proceedings of the National Academy of Sciences, 2014. **111**(23): p. 8619.
13. Ponce, C.R., et al., *Evolving Images for Visual Neurons Using a Deep Generative Network Reveals Coding Principles and Neuronal Preferences*. Cell, 2019. **177**(4): p. 999-1009.e10.
14. Chang, L. and D.Y. Tsao, *The Code for Facial Identity in the Primate Brain*. Cell, 2017. **169**(6): p. 1013-1028.e14.
15. Cao, R., et al., *Feature-based encoding of face identity by single neurons in the human medial temporal lobe*. bioRxiv, 2020: p. 2020.09.01.278283.
16. Rutishauser, U., et al., *Representation of retrieval confidence by single neurons in the human medial temporal lobe*. Nat Neurosci, 2015. **18**(7): p. 1041-1050.
17. Quian Quiroga, R., et al., *Invariant visual representation by single neurons in the human brain*. Nature, 2005. **435**(7045): p. 1102-1107.
18. Quian Quiroga, R., *Concept cells: the building blocks of declarative memory functions*. Nature Reviews Neuroscience, 2012. **13**: p. 587.
19. Rey, H.G., et al., *Encoding of long-term associations through neural unitization in the human medial temporal lobe*. Nature Communications, 2018. **9**(1): p. 4372.
20. Kreiman, G., C. Koch, and I. Fried, *Category-specific visual responses of single neurons in the human medial temporal lobe*. Nat Neurosci, 2000. **3**(9): p. 946-953.

21. Rutishauser, U., A.N. Mamelak, and E.M. Schuman, *Single-Trial Learning of Novel Stimuli by Individual Neurons of the Human Hippocampus-Amygdala Complex*. *Neuron*, 2006. **49**(6): p. 805-813.
22. Rutishauser, U., et al., *Human memory strength is predicted by theta-frequency phase-locking of single neurons*. *Nature*, 2010. **464**(7290): p. 903-907.
23. Loffler, G., et al., *fMRI evidence for the neural representation of faces*. *Nat Neurosci*, 2005. **8**(10): p. 1386-1391.
24. Carlin, J.D. and N. Kriegeskorte, *Adjudicating between face-coding models with individual-face fMRI responses*. *PLOS Computational Biology*, 2017. **13**(7): p. e1005604.
25. Bashivan, P., K. Kar, and J.J. DiCarlo, *Neural population control via deep image synthesis*. *Science*, 2019. **364**(6439): p. eaav9436.
26. Cao, R., et al., *A Flexible Neural Representation of Faces in the Human Brain*. *Cerebral Cortex Communications*, 2020. **1**(1): p. tgaa055.
27. Rutishauser, U., *Testing Models of Human Declarative Memory at the Single-Neuron Level*. *Trends in Cognitive Sciences*, 2019. **23**(6): p. 510-524.
28. Quian Quiroga, R., et al., *Sparse but not 'Grandmother-cell' coding in the medial temporal lobe*. *Trends in Cognitive Sciences*, 2008. **12**(3): p. 87-91.
29. Rutishauser, U., et al., *The Architecture of Human Memory: Insights from Human Single-Neuron Recordings*. *The Journal of Neuroscience*, 2021. **41**(5): p. 883.
30. Fried, I., et al., *Single Neuron Studies of the Human Brain: Probing Cognition*. 2014, Boston: MIT Press.
31. Mormann, F., et al., *A category-specific response to animals in the right human amygdala*. *Nat Neurosci*, 2011. **14**(10): p. 1247-1249.
32. Faraut, M.C.M., et al., *Dataset of human medial temporal lobe single neuron activity during declarative memory encoding and recognition*. *Scientific Data*, 2018. **5**(1): p. 180010.
33. Donoghue, T., et al., *Single neurons in the human medial temporal lobe flexibly shift representations across spatial and memory tasks*. *Hippocampus*, 2023. **33**(5): p. 600-615.
34. Reber, T.P., et al., *Representation of abstract semantic knowledge in populations of human single neurons in the medial temporal lobe*. *PLOS Biology*, 2019. **17**(6): p. e3000290.
35. Cao, R., et al., *A human single-neuron dataset for face perception*. *Scientific Data*, 2022. **9**(1): p. 365.
36. Kyzar, M., et al., *Dataset of human-single neuron activity during a Sternberg working memory task*. *Scientific Data*, 2024. **11**(1): p. 89.
37. Rutishauser, U., et al., *Representation of retrieval confidence by single neurons in the human medial temporal lobe*. *Nat Neurosci*, 2015. **18**(7): p. 1041-50.

Reply to comments from Reviewer 1

The authors have made a good attempt to address all my concerns. I have no further questions.

Once again, we thank the reviewer for the expert and constructive comments.

Reply to comments from Reviewer 2

The response arguments are mostly convincing and I appreciate that the authors tempered their claims based on the acknowledged limitations. I will note however that the suggestion that the comparison of remembered to forgotten items in the memory test constitutes an experimental manipulation/condition is misleading, as behavior assigns trials to these conditions, not the experimenters, which is non-random selection. Extraneous factors like attentional lapses, drowsiness, or even symptoms related to the patients' neurological condition or intervention could be driving these differences.

I think the results and main conclusions are now mostly convincing, but am still of the opinion that the findings are very narrow: properties of a very small subset of hippocampal neurons that provide modest correlational evidence in favor of one response model over another, without very thorough consideration of relationship to memory more broadly. I'm skeptical that this will have much impact on memory or visual perception fields..

Once again, we thank the reviewer for the expert and constructive comments. We have further included the following discussion in the revised manuscript:

“Lastly, we acknowledge that in our recognition memory tasks, the classification of remembered vs. forgotten items is inherently behavior-driven rather than experimentally manipulated, which may introduce potential confounds such as attentional lapses or patient-specific factors. We also recognize that our study focuses on a specific subset of MTL neurons and provides correlational rather than causal evidence for the proposed response model. Nonetheless, our findings represent an important initial step toward understanding how MTL neurons transform visual features into memories, highlighting a critical link between visual feature coding and memory formation. We believe our work will inspire further research to explore these mechanisms more comprehensively using causal methodologies.”

Reply to comments from Reviewer 3

I have no further comments.

Once again, we thank the reviewer for the expert and constructive comments.